



Earth System
Science
Data

# Global variability in belowground autotrophic respiration in terrestrial ecosystems

**Xiaolu Tang[1,2], Shaohui Fan[3], Wenjie Zhang[4,5], Sicong Gao[5], Guo Chen[1], and Leilei Shi[6]**

[1]College of Earth Science, Chengdu University of Technology, Chengdu, China
[2]State Environmental Protection Key Laboratory of Synergetic Control and Joint Remediation for Soil & Water Pollution, Chengdu University of Technology, Chengdu, China
[3]Key Laboratory of Bamboo and Rattan, International Centre for Bamboo and Rattan, Beijing, China
[4]State Key Laboratory of Resources and Environmental Information System,
Institute of Geographic Sciences and Natural Resources Research, Beijing, China
[5]School of Life Sciences, University of Technology Sydney, Sydney, New South Wales, Australia
[6]Key Laboratory of Geospatial Technology for the Middle and Lower Yellow River Regions,
College of Environment and Planning, Henan University, Jinming Avenue, Kaifeng, China

**Correspondence:** Wenjie Zhang (wenjie.zhang@uts.edu.au) and Sicong Gao
(sicong.gao@student.uts.edu.au)

**Abstract.** Belowground autotrophic respiration (RA) is one of the largest but most highly uncertain carbon flux components in terrestrial ecosystems. However, RA has not been explored globally before and still acts as a "black box" in global carbon cycling currently. Such progress and uncertainty motivate the development of a global RA dataset and understanding its spatial and temporal patterns, causes, and responses to future climate change. We applied the random forest (RF) algorithm to upscale an updated dataset from the Global Soil Respiration Database (v4) – covering all major ecosystem types and climate zones with 449 field observations, using globally gridded temperature, precipitation, soil and other environmental variables. We used a 10-fold cross validation to evaluate the performance of RF in predicting the spatial and temporal pattern of RA. Finally, a globally gridded RA dataset from 1980 to 2012 was produced with a spatial resolution of $0.5° \times 0.5°$ (longitude $\times$ latitude) and a temporal resolution of 1 year (expressed in $g\,C\,m^{-2}\,yr^{-1}$; grams of carbon per square meter per year).

Globally, mean RA was $43.8 \pm 0.4\,Pg\,C\,yr^{-1}$, with a temporally increasing trend of $0.025 \pm 0.006\,Pg\,C\,yr^{-2}$ from 1980 to 2012. Such an incremental trend was widespread, representing 58 % of global land. For each 1 °C increase in annual mean temperature, global RA increased by $0.85 \pm 0.13\,Pg\,C\,yr^{-2}$, and it was $0.17 \pm 0.03\,Pg\,C\,yr^{-2}$ for a 10 mm increase in annual mean precipitation, indicating positive feedback of RA to future climate change. Precipitation was the main dominant climatic driver controlling RA, accounting for 56 % of global land, and was the most widely spread globally, particularly in dry or semi-arid areas, followed by shortwave radiation (25 %) and temperature (19 %). Different temporal patterns for varying climate zones and biomes indicated uneven responses of RA to future climate change, challenging the perspective that the parameters of global carbon stimulation are independent of climate zones and biomes. The developed RA dataset, the missing carbon flux component that is not constrained and validated in terrestrial ecosystem models and Earth system models, will provide insights into understanding mechanisms underlying the spatial and temporal variability in belowground vegetation carbon dynamics. The developed RA dataset also has great potential to serve as a benchmark for future data–model comparisons. The developed RA dataset in a common NetCDF format is freely available at https://doi.org/10.6084/m9.figshare.7636193 (Tang et al., 2019).

## 1   Introduction

Belowground autotrophic respiration (RA) mainly originated from plant roots, mycorrhizae, and other microorganisms in the rhizosphere directly relying on the labile carbon component leaked from roots (Hanson et al., 2000; Tang et al., 2016; Wang and Yang, 2007). Thus, RA reflects the photosynthesis-derived carbon respired back to the atmosphere by roots and regulates the net photosynthetic production allocation to belowground tissues (Högberg et al., 2002). RA is one main component of soil respiration (Hanson et al., 2000), and soil respiration represents the second largest source of carbon fluxes from soil to the atmosphere (after gross primary production – GPP) in the global carbon cycle (Raich and Schlesinger, 1992). Globally, RA could amount to roughly $54 \, \mathrm{Pg \, C \, yr^{-1}}$ ($1 \, \mathrm{Pg} = 10^{15} \, \mathrm{g}$, calculating RA as an approximate ratio of 0.5 of soil respiration; more details in Hanson et al., 2000) according to different estimates of global soil respiration (Bond-Lamberty, 2018), which is almost 5 times the carbon release from human activities (Le Quéré et al., 2018). However, the contribution of RA to soil respiration varied greatly, from 10 % to 90 % across biomes, across climate zones and among years (Hanson et al., 2000), leading to the strong spatial and temporal variability in RA. Thus, whether RA varies with ecosystem types or climate zones remains an open question at the global scale (Ballantyne et al., 2017). Consequently, an accurate estimate of RA and its spatio-temporal dynamics are critical in understanding the response of terrestrial ecosystems to climate change.

Due to the difficulties in separation and measurement of RA at varying spatial scales and its diurnal, seasonal and annual variabilities, RA becomes one of the largest but most highly uncertain carbon flux components in terrestrial ecosystems. Although individual site measurements of RA have been widely conducted across ecosystem types and biomes, the globally spatial and temporal patterns of RA have not been explored and still act as a black box in global carbon cycling (Ballantyne et al., 2017). This black box is not well constrained and validated because most terrestrial ecosystem models and Earth system models were commonly calibrated and validated against eddy covariance measurements of net ecosystem carbon exchange (Yang et al., 2013). Such progress and uncertainty motivate the development of a global RA dataset from observations and understanding its spatial and temporal patterns, causes, and responses to future climate change. Despite the general agreement that global soil respiration increased during last several decades (Bond-Lamberty et al., 2018; Bond-Lamberty and Thomson, 2010; Zhao et al., 2017), how global RA responds to climate change is far from certain because of different temperature sensitivities of RA across terrestrial ecosystems (Liu et al., 2016; Wang et al., 2014). Therefore, reducing RA uncertainty and clarifying its response to climate change, particularly to temperature and precipitation, is essential for global carbon allocation and future projection of the impact of climate change on global terrestrial carbon cycling.

Although several studies have globally estimated soil respiration and its response to climate variables (Bond-Lamberty and Thomson, 2010; Hursh et al., 2017; Zhao et al., 2017), such efforts have not been conducted for global RA directly. Hashimoto et al. (2015) indirectly derived RA via the difference between total soil respiration and heterotrophic respiration; however, it might lead to uncertainties due to the inclusion of the temperature and precipitation as the only model drivers and a low model efficiency (32 %). Besides temperature and precipitation, other variables, e.g., soil water, carbon and nitrogen content, are additionally critical factors regulating RA, and those factors generally varied with biomes and climate zones. Consequently, Hashimoto et al. (2015) may not reflect the key processes affecting RA, such as soil nutrient constraints.

On the other hand, the climate-derived models usually explain $< 50 \%$ variability in soil respiration (Bond-Lamberty and Thomson, 2010; Hashimoto et al., 2015; Hursh et al., 2017), which might be another uncertainty source. Recent studies have included more variables and field observations to improve the prediction ability of linear and nonlinear models (Jian et al., 2018b; Zhao et al., 2017); however, this may propagate error because of the overfitting and autocorrelation among these variables (Long and Freese, 2006). The random forest (RF; Breiman, 2001) algorithm, a machine-learning approach, could overcome these issues based on the hierarchical structure and be insensitive to outliers and noise compared to single classifiers (Breiman, 2001; Tian et al., 2017). RF uses a large number of ensemble regression trees but a random selection of predictive variables (Breiman, 2001). RF only requires two free parameter settings: the number of variables sampled as candidates for each split and the number of trees. The performance of the RF model is usually not sensitive to the number of trees and number of variables. Moreover, RF regression can deal with a large number of features, which could help feature selection based on the variable importance and can avoid overfitting (Jian et al., 2018b). Consequently, it has been widely used for carbon flux modeling in recent years (Bodesheim et al., 2018; Jung et al., 2017).

Therefore, we applied the RF algorithm to retrieve global RA based on the updated RA field observations from the most updated Global Soil Respiration Database (SRDB v4; Bond-Lamberty and Thomson, 2018) with the linkage of other global variables (see Materials and methods) for the first time, aiming to (1) develop a globally gridded RA dataset using field observations (named RF-RA), (2) estimate the spatial and temporal patterns of RA at the global scale, (3) identify the dominant driving factors of the spatial and temporal variabilities in RA, and (4) compare the RF-RA dataset with the previous RA estimates from Hashimoto

et al. (2015). The developed RF-RA dataset will advance our understanding of global RA and its spatial and temporal variabilities. The RF-RA is expected to serve as a benchmark for global vegetation models and future data–model comparison, which further advance our knowledge of the covariation of RA with climate, soil and vegetation factors, linking the empirical observations temporally and spatially to bridge the knowledge gap on local, regional and global scales.

## 2 Material and methods

### 2.1 Development of RA observational dataset

First, the RA observational dataset was developed based on SRDB (v4) across the globe, which is publicly available at https://daac.ornl.gov/cgi-bin/dsviewer.pl?ds_id=1578 (last access: 18 November 2018; Bond-Lamberty and Thomson, 2018). Then, we further updated the dataset using observations collected from Chinese peer-reviewed literature from the China Knowledge Resource Integrated Database (CNKI: https://www.cnki.net/, last access: 1 December 2017) up to March 2018, which followed the identical criteria applied in SRDB development. To control the data quality, annual RA observations were filtered in that (1) annual RA was directly reported in publications indicated by the "years of data" of SRDB; (2) the start and end years were recorded in literature or expanded from the years of data of SRDB; (3) soil respiration measurements with alkali absorption and soda lime were not included due to the potential underestimate of respiration rate with the increasing pressure inside chamber (Pumpanen et al., 2004); (4) observations with treatments of nitrogen addition, air and soil warming, and rain and litter exclusion were not included, except cropland; and (5) potential problems observations labeled as Q10 (potential problem with data), Q11 (suspected problem with data), Q12 (known problem with data), Q13 (duplicate) and Q14 (inconsistency) were excluded. Finally, this study included a total of 449 field observations (Fig. 1), including 68 observations from CNKI. RA observations were absolutely dominated by forest ecosystems (379 observations) that are globally unevenly distributed, mainly from China, America and Europe. Although there was a lack of RA observations in Australia, Russia, Africa, and South America, our dataset covered all major ecosystem types and climate zones across the globe.

### 2.2 Vegetation, climate and soil data

A total of 11 environmental variables were used to model global RA (Table 1). Specifically, global land cover with a half-degree resolution was obtained from MODIS Land Cover (MCD12Q1 v5; Friedl et al., 2010). The monthly gridded temperature, precipitation, diurnal temperature range, potential evapotranspiration and self-calibrated Palmer drought severity index (PDSI) at the 0.5° resolution were obtained from Climatic Research Unit (CRU) Time Se-

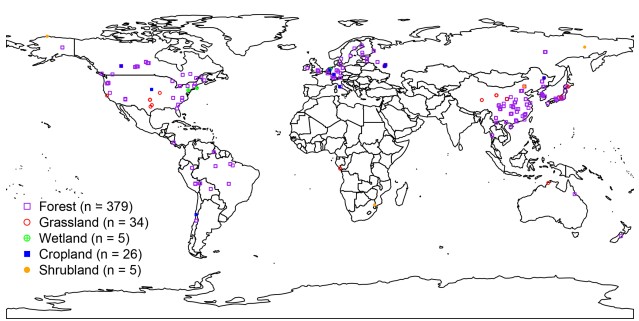

**Figure 1.** Distribution of observational sites used to develop the globally gridded RF-RA dataset.

ries (TS) Version 4.01 from 1901 to 2016 (Harris et al., 2014; van der Schrier et al., 2013). Monthly shortwave radiation (SWR; Kalnay et al., 1996) and soil water content (van den Dool et al., 2003) at the 0.5° resolution were from the National Oceanic and Atmospheric Administration Earth System Research Laboratory (NOAA ESRL) Physical Sciences Division. Soil organic carbon content with a resolution of 250 m was downloaded from soil grid data (Hengl et al., 2017), and soil nitrogen density was from the Global Soil Data Task of the International Geosphere-Biosphere Programme (IGBP; Global Soil Data, 2000), while monthly nitrogen deposition data with a resolution of 0.5° were downloaded from the Earth system models of GISS-E2-R, CCSM-CAM3.5 and GFDL-AM3, providing coverage since 1850s (Lamarque et al., 2013). The monthly global variables were first aggregated to the year scale and then resampled to a 0.5° resolution using bilinear interpolation for those variables without a 0.5° resolution. These variables could represent different aspects controlling RA variability. For instance, temperature, precipitation and soil water content are the most important variables controlling plant photosynthesis, which is the primary carbon source of RA (Högberg et al., 2002, 2001). Finally, global variables of each given site extracted by coordinates correspond to annual RA estimates from the SRDB.

### 2.3 Random forest-based RA modeling

A RF model was trained with the 11 variables listed in Table 1 by *caret* by linking *RandomForest* package in R 3.4.4 (Kabacoff, 2015); then the trained model was implemented to estimate grid RA at the 0.5° × 0.5° resolution over 1980–2012. The performance of RF was assessed by a 10-fold cross validation (CV). A 10-fold CV suggested that the whole dataset was subdivided into 10 parts with an approximately equal number of samples. The target values for each of these 10 parts were predicted on the training using the remaining nine parts. Two statistics were employed in model assessment: modeling efficiency ($R^2$) and root-mean-square error (RMSE; Yao et al., 2018). The 10-fold CV result

**Table 1.** Global variables used for producing the global RH dataset.

| | Variables | Type | Type of variability | Sources |
|---|---|---|---|---|
| Climate | Mean annual temperature (°C) | Split | Yearly | https://crudata.uea.ac.uk/cru/data/hrg/cru_ts_4.01/ |
| | Mean annual precipitation (mm) | Split | Yearly | (last access: September 2017; Harris et al., 2014) |
| | Diurnal temperature range (°C) | Split | Yearly | https://crudata.uea.ac.uk/cru/data/drought/ |
| | Potential evapotranspiration (mm) | Split | Yearly | (last access: October 2017; van der Schrier et al., 2013) |
| | Palmer drought severity index | Split | Yearly | |
| | Nitrogen deposition ($\mathrm{g\,N\,m^{-2}\,yr^{-1}}$) | Split | Yearly | https://www.isimip.org/gettingstarted/ availability-input-data-isimip2b/ (last access: August 2013; Lamarque et al., 2013) |
| | Downward shortwave radiation ($\mathrm{W\,m^{-2}}$) | Split | Yearly | ftp://ftp.cdc.noaa.gov/Datasets/ncep.reanalysis.derived/ surface_gauss/dswrf.sfc.mon.mean.nc (last access: October 2019; Kalnay et al., 1996) |
| Soil | Soil carbon content ($\mathrm{g\,kg^{-1}}$) | – | Static | https://soilgrids.org/#!/?layer=TAXNWRB_250m (last access: June 2016; Hengl et al., 2017) |
| | Soil nitrogen density ($\mathrm{g\,m^{-2}}$) | – | Static | https://webmap.ornl.gov/ogc/dataset.jsp?ds_id=569 (last access: 24 August 2018; Global Soil Data, 2000) |
| | Soil water content (mm) | Split | Yearly | https://www.esrl.noaa.gov/psd/data/gridded/data.cpcsoil. html (last access: September 2019; van den Dool et al., 2003) |
| Land cover | MODIS land cover | – | Static | https://search.earthdata.nasa.gov/search/granules?p= C203669657-LPDAAC_ECS&q=modislandcover&tl= 1556377353!4!! (last access: July 2019; Friedl et al., 2010) |

showed that RF performed well and could capture the spatial and temporal pattern of RA (Fig. S1 in the Supplement).

## 2.4 Temporal trend analysis

We applied Theil–Sen linear regression to estimate temporal trend analysis of RA and its driving variables for each grid cell. The Theil–Sen estimator is a median-based nonparametric slope estimator which has been widely used for spatial analysis of time series carbon flux analysis (Forkel et al., 2016; Zhang et al., 2017). The Mann–Kendall nonparametric test was applied for the significant change trend in RA and its driving factors for each grid cell ($p < 0.05$).

## 2.5 Relationships between RA and climate variables

Mean annual temperature, precipitation and shortwave radiation were considered to be the most important proxies driving RA. The relationships between RA and temperature, precipitation and shortwave radiation were analyzed by partial correlation for each grid cell. The absolute value of the correlation coefficient of these three variables was used in an RGB (red, green, blue) combination to indicate the dominant factors of RA.

## 2.6 Cross comparisons with Hashimoto2015-RA

To further compare the differences between the RF-RA dataset and RA developed by Hashimoto et al. (2015; named Hashimoto2015-RA), the comparison map profile (CMP) method was applied. Hashimoto developed a climate-driven model by updating Raich's model, which stimulated soil respiration as a function of temperature and water (precipitation) at a monthly time step (Hashimoto et al., 2015; Raich et al., 2002). Therefore, to get a global estimate to soil respiration at a monthly scale, the globally gridded air temperature and precipitation with a spatial resolution of 0.5° were derived from University of East Anglia CRU 3.21 (Harris et al., 2014), and 1638 field observations were taken from the SRDB (v3) for the model parameterization (Hashimoto et al., 2015). Monthly soil respiration was summed to a yearly scale. Furthermore, annual soil respiration was divided into autotrophic and heterotrophic respiration using a global relationship between soil respiration and heterotrophic respiration derived from a meta-analysis (Bond-Lamberty et al., 2004). This global relationship can be expressed as

$$\ln(\mathrm{RH}) = 1.22 + 1.73 \times \ln(\mathrm{RS}), \tag{1}$$

TS1 where RH means annual heterotrophic respiration, and RS stands for annual soil respiration (expressed in $\mathrm{g\,C\,m^{-2}\,yr^{-1}}$).

Therefore, global Hashimoto2015-RA was derived by the difference between soil respiration and heterotrophic respiration. The monthly or annual Hashimoto2015-RA dataset can be freely accessed from (http://cse.ffpri.affrc.go.jp/shojih/data/index.html, last access: February 2016; Hashimoto et al., 2015).

The CMP was developed based on absolute distance ($D$) and the cross-correlation coefficient (CC) on multiple scales (Gaucherel et al., 2008). $D$ and CC reflect the similarity of data values and spatial structure of two images with the same size, respectively (Gaucherel et al., 2008). Low $D$ and higher CC reflect goodness between the compared images, while high $D$ and low CC suggest badness. The $D$ among moving windows of two compared images was calculated by Eq. (2) (Gaucherel et al., 2008):

$$D = \mathrm{abs}(\overline{x} - \overline{y}). \tag{2}$$

$\overline{x}$ and $\overline{y}$ are averages calculated over two moving windows (3 pixels×3 pixels to 41 pixels×41 pixels in this study). Finally, the mean $D$ was averaged for different scales.

The CC was calculated by Eq. (3) (Gaucherel et al., 2008):

$$CC = \frac{1}{N^2} \sum_{i=1}^{N} \sum_{j=1}^{N} \frac{(x_{ij} - \overline{x}) \times (y_{ij} - \overline{y})}{\sigma_x \times \sigma_y}, \tag{3}$$

with $\quad \sigma_x^2 = \dfrac{1}{N^2 - 1} \sum_{i=1}^{N} \sum_{j=1}^{N} (x_{ij} - \overline{x})^2, \tag{4}$

where $x_{ij}$ and $y_{ij}$ are the pixel values at row $i$ and column $j$ of two moving windows of the two compared images, respectively. $N$ represents the number of pixels for each moving window, while $\sigma_x$ and $\sigma_y$ are the standard deviation calculated from the two moving windows. Finally, like $D$ calculations, CC was calculated as the mean of different scales.

## 3 Results

### 3.1 Spatial patterns of RA

The RF-RA dataset presented a great globally spatial variability during 1980–2012 (Figs. 2a and 3). The largest RA fluxes commenced from tropical regions, particularly in the tropical Amazon and southeastern areas, which generally have a high RA that is more than $700\,\mathrm{g\,C\,m^{-2}\,yr^{-1}}$ (grams of carbon per square meter per year). Following the tropical areas, the subtropics, e.g., southern China and eastern America, and humid temperate areas, e.g., North America and western and central Europe, had typical moderate RA fluxes of 400–$600\,\mathrm{g\,C\,m^{-2}\,yr^{-1}}$. By contrast, the relative low RA fluxes occurred in the areas with sparse vegetation cover and cold and dry climate, e.g., boreal and tundra, which had low temperatures and a short growing season. Besides this, dry or semi-arid areas, e.g., northwestern China and the Middle East, also

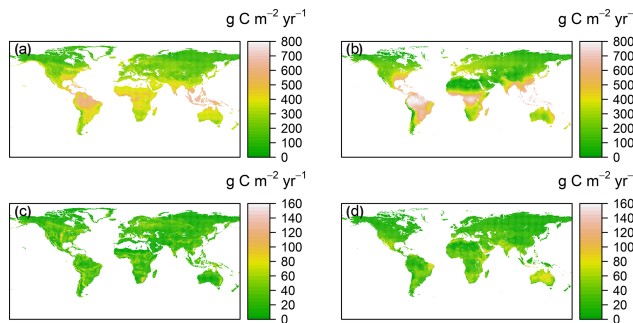

**Figure 2.** Spatial patterns of annual mean and standard deviation for RF-RA (**a, c**) and Hashimoto2015-RA (**b, d**) from 1980 to 2012. The standard deviation was applied to characterize the inter-annual variability following Yao et al. (2018).

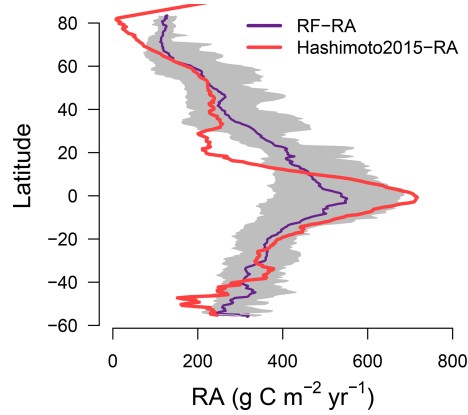

**Figure 3.** Latitudinal pattern for RF-RA and Hashimoto2015-RA. The grey area indicates 2.5th to 97.5th percentile ranges of the RF-RA.

had typical low RA fluxes below $200\,\mathrm{g\,C\,m^{-2}\,yr^{-1}}$, which were often limited by water availability.

The most significant RA inter-annual variability (expressed as standard deviation; Fig. 2c) was found in topical or subtropical regions, with values above $80\,\mathrm{g\,C\,m^{-2}\,yr^{-1}}$, while most areas remained less variable, with values less than $40\,\mathrm{g\,C\,m^{-2}\,yr^{-1}}$. Latitudinally, zonal mean RA increased from cold and dry biomes (tundra and semi-arid) to warm and humid biomes (temperate and tropical forest; Fig. 3), reflecting more to fewer environmental limitations. RA varied from $112 \pm 21\,\mathrm{g\,C\,m^{-2}\,yr^{-1}}$ at about $70°\,\mathrm{N}$ to $552 \pm 101\,\mathrm{g\,C\,m^{-2}\,yr^{-1}}$ at the Equator. Between 10–25° S and 15–20° N, due to the limitation of water, zonal mean RA experienced a slight decrease. Therefore, with the increase in water availability, RA led to a second peak in around 20° N and 40° S, respectively.

Compared to RF-RA, Hashimoto2015-RA presented a similarly latitudinal pattern, with the highest RA fluxes in tropical regions characterized by warm and humid climate, followed by subtropical regions and the lowest RA in boreal areas featured by cold and dry climate (Fig. 2b). The

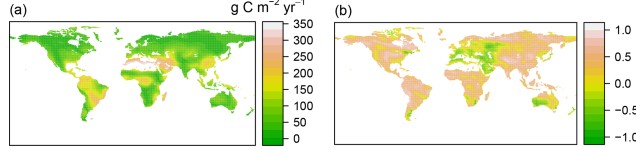

**Figure 4.** Comparison of RF-RA with Hashimoto2015-RA based on absolute distance **(a)** and cross correlation **(b)**.

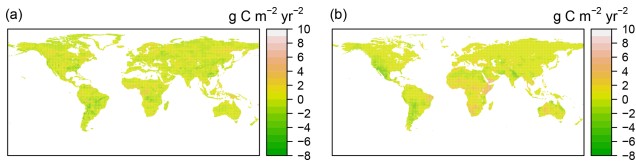

**Figure 5.** Spatial patterns of the temporal trend for RF-RA and Hashimoto2015-RA during 1980–2012.

most significant change occurred in tropical areas and central Australia. However, it is worth noting that some clear differences between data-derived RA and Hashimoto2015-RA existed (Fig. 4): specifically, there was a remarkable difference of above $300 \, \mathrm{g \, C \, m^{-2} \, yr^{-1}}$ for the southern Amazon and $200 \, \mathrm{g \, C \, m^{-2} \, yr^{-1}}$ for subtropical China. Although most areas between RF-RA and Hashimoto2015-RA expressed high and positive correlations, some areas, such as the Middle East, western Russia, eastern America and northern Japan, showed negative correlations.

### 3.2 Spatial pattern of RA trend

The trend of RF-RA showed heterogeneous spatial patterns (Fig. 5). A total of 58 % of global areas experienced an increasing trend during 1980–2012 (calculating from cell areas), and 33 % of these areas showed a significant change ($p < 0.05$). Generally, the change trend for the majority areas was from $-4$ to $4 \, \mathrm{g \, C \, m^{-2} \, yr^{-2}}$, while the most striking increasing change occurred in eastern Russia and tropical and eastern regions in Africa, with an increasing trend of above $5 \, \mathrm{g \, C \, m^{-2} \, yr^{-2}}$. Similarly, 77 % of global areas of Hashimoto2015-RA had an increasing trend, 46 % of which were statistically significant ($p < 0.05$).

### 3.3 Total RA and its temporal trend

Mean global RA was $43.8 \pm 0.4 \, \mathrm{Pg \, C \, yr^{-1}}$ during 1980–2012, varying from $42.9 \, \mathrm{Pg \, C \, yr^{-1}}$ in 1992 to $44.9 \, \mathrm{Pg \, C \, yr^{-1}}$ in 2010, with a significant trend of $0.025 \pm 0.006 \, \mathrm{Pg \, C \, yr^{-2}}$ despite high annual variabilities ($0.06 \, \% \, \mathrm{yr^{-1}}$, $p < 0.001$; Fig. 6a). Similarly, a rising trend was also observed for Hashimoto2015-RA ($0.073 \pm 0.009 \, \mathrm{Pg \, C \, yr^{-2}}$, $p < 0.001$; Fig. 6b), which was higher than that of RF-RA. The annual mean of Hashimoto2015-RA was $40.5 \pm 0.9 \, \mathrm{Pg \, C \, yr^{-1}}$.

RA and its trend were also evaluated for three climate zones (boreal, temporal and tropical areas based on

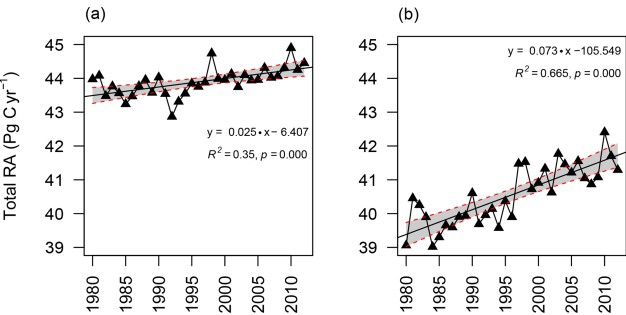

**Figure 6.** Annual variability in RF-RA **(a)** and Hashimoto2015-RA **(b)** from 1980 to 2012. The grey area represents 95 % confidence interval.

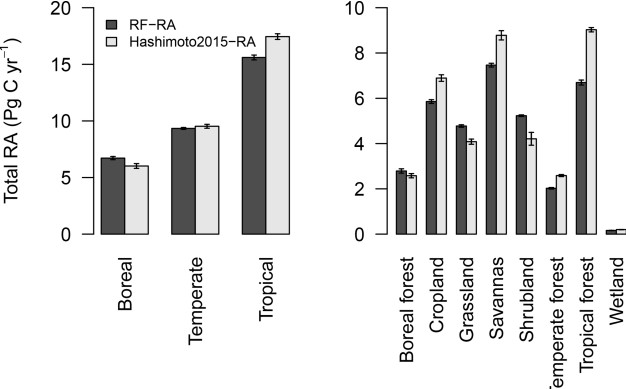

**Figure 7.** Total amount of RF-RA and Hashimoto2015-RA for three climate zones and eight biomes during 1980–2012. Three climate zones defined as boreal, temperature and tropical regions according to Peel et al. (2007), while eight biomes include boreal forest, cropland, grassland, savannas, shrubland, temperate forest, tropical forest and wetland. The error bars indicated standard deviation.

the Köppen–Geiger climate classification) and eight major biomes (boreal forest, cropland, grassland, savannas, shrubland, temperate forest, tropical forest and wetland; Fig. 7). The tropics had the highest RA, $15.6 \pm 0.2 \, \mathrm{Pg \, C \, yr^{-1}}$, followed by temperate regions, with $9.3 \pm 0.1 \, \mathrm{Pg \, C \, yr^{-1}}$, and boreal areas represented the lowest RA, $6.7 \pm 0.1 \, \mathrm{Pg \, C \, yr^{-1}}$. These three climate zones were the main contributors of global RA, accounting for 72 %. Temporally, considerable RA inter-annual variability in these three climate zones existed (Fig. S2). Specifically, RA in tropical and boreal zones showed a significantly increasing trend from 1980 to 2012, with an increasing rate of $0.013 \pm 0.003$ and $0.008 \pm 0.002 \, \mathrm{Pg \, C \, yr^{-2}}$, respectively. However, RA in temperate zones presented a slightly decreasing trend of $-0.003 \pm 0.001 \, \mathrm{Pg \, C \, yr^{-2}}$ ($p = 0.048$), although strong variability was observed.

In terms of biomes, tropical forest had the highest RA, followed by the widely distributed cropland and savannas (Fig. 7), while wetland had the lowest RA due to its lim-

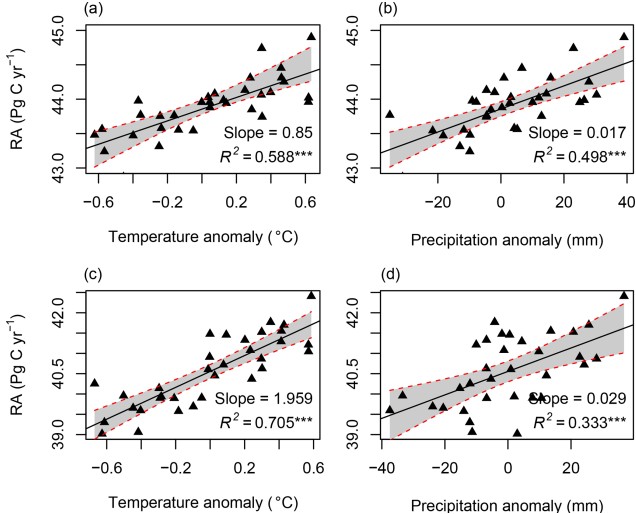

**Figure 8.** The correlation between RA and temperature and precipitation: **(a, b)** for RF-RA and **(c, d)** for Hashimoto2015-RA. The anomaly was calculated as the difference between temperature or precipitation of corresponding year and the mean of 1980–2012. *** means significant level at 0.001.

ited land cover. RA showed a significantly increasing trend during 1980–2012 ($p_s < 0.01$) in the majority of biomes, except for temperate forest, savannas and wetland. RA in tropical forest, boreal forest and cropland increased by $0.0076 \pm 0.0015$, $0.0047 \pm 0.0016$ and $0.0036 \pm 0.0014$ Pg C yr$^{-2}$, respectively. Compared to RF-RA, Hashimoto2015-RA for the three climate zones and eight biomes generally produced similar change patterns, although the magnitude difference existed (Figs. 7, S2 and S3). However, there were significant increasing trends of total RA in temperate zones, temperate forest, savannas and wetland of Hashimoto2015-RA which were not observed in RF-RA.

RA was significantly correlated with the temperature anomaly ($R^2 = 0.59$, $p < 0.001$) and precipitation anomaly ($R^2 = 0.50$, $p < 0.001$; Fig. 8). On average, RA increased by $0.85 \pm 0.13$ Pg C yr$^{-2}$ for a 1 °C increment in mean annual temperature and $0.17 \pm 0.03$ Pg C yr$^{-2}$ for a 10 mm increase in mean annual precipitation. However, different biomes and climate zones showed uneven responses to the temperature and precipitation changes (Figs. S4 and S5). For example, no significant correlations were found between RA in the temperate zone, savannas, wetland and the temperature anomaly, while other climate zones and biomes were significantly correlated with the temperature and precipitation anomaly.

## 4   Dominant factors for RA variability

The dominant environmental factor was examined with partial regression coefficients when regressing RA against annual mean temperature, precipitation and shortwave radiation. Latitudinally, higher mean annual temperature, precip-

itation and shortwave radiation were associated with higher RA in the major latitudinal gradients (positive partial correlations; Fig. S6). Spatially, the dominant environmental factor varied greatly globally (Fig. 9). Precipitation was the most important dominant factor for the spatial pattern of RA among the three environmental controls, covering about 56 % of global land (Fig. 10), and it was widely distributed globally, particularly in dry or semi-arid areas such as northwestern China, southern Africa, central Australia and America. Temperature dominated about 19 % of global land, which mainly occurred in tropical Africa, southern Amazon rainforest and Siberia, and partly in the tundra. The rest of the land (25 %) was dominated by shortwave radiation, primarily covering boreal areas above 50° N in eastern America and central and eastern Russia. Similarly, precipitation was also the most important dominant factor for Hashimoto2015-RA, dominating about 77 % of land, while temperature and shortwave radiation dominated 13 % and 10 % of land. However, their spatial patterns varied greatly compared to RF-RA. For example, temperature was the main dominant factor for most areas in Australia for Hashimoto2015-RA, while RF-RA indicated that precipitation and shortwave radiation dominated such areas (Fig. 9).

## 5   Discussion

### 5.1   Global RA

Despite great efforts to quantify global soil carbon fluxes and their spatial and temporal patterns (Bond-Lamberty and Thomson, 2010; Hursh et al., 2017; Jian et al., 2018b), to our knowledge, no attempt tried to assess RA using the machine-learning approach by linking a large number of empirical measurements, and RA's spatial and temporal patterns remain large uncertainties. Such uncertainties justify the development of a global RA dataset derived from observations to understand its spatial and temporal patterns, causes, and responses to future climate change. Based on the most updated observations from the SRDB (Bond-Lamberty and Thomson, 2018) and Chinese peer-reviewed literature, we, for the first time, applied the RF algorithm to develop the RF-RA dataset and estimate the temporal and spatial variability in global RA and its response to environmental variables, which can indeed contribute to reduce RA uncertainties.

Globally, mean annual RA amounted to $43.8 \pm 0.4$ Pg C yr$^{-1}$ from 1980 to 2012 (Fig. 6). It was slightly higher than Hashimoto2015-RA ($40.5 \pm 0.9$ Pg C yr$^{-1}$), and there was great divergence of spatial and temporal patterns (see discussion part in Comparison with Hashimoto2015-RA). Due to there being no direct estimate on global RA, the RF-RA dataset was compared with other RA estimates using total soil respiration multiplied by the proportion of RA or heterotrophic respiration. The global average proportion of RA ranged from 0.37 to 0.46 over 1990–2014 (calculated from Bond-Lamberty et al., 2018), while

(a) 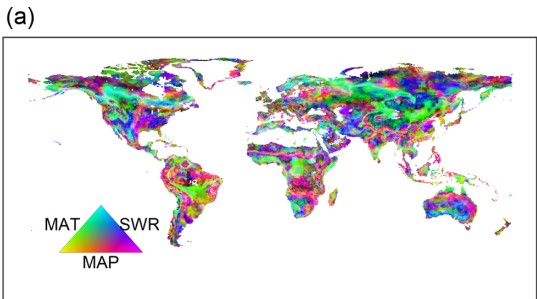        (b) 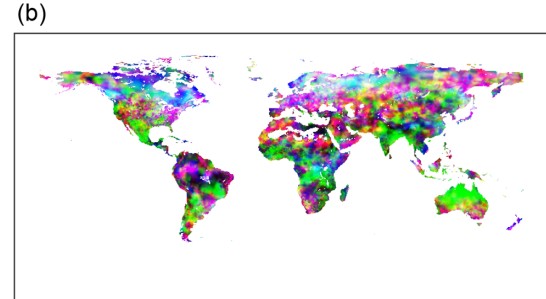

**Figure 9.** Dominant driving factors for RF-RA **(a)** and Hashimoto2015-RA **(b)**. MAT is mean annual temperature, MAP is mean annual precipitation and SWR is shortwave radiation.

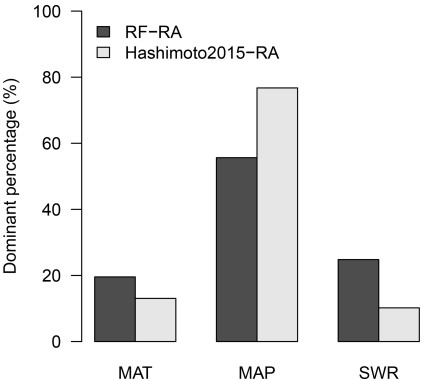

**Figure 10.** The percentage of land (calculated from cell areas) dominated by mean annual temperature (MAT), precipitation (MAP) and shortwave radiation (SWR) for RF-RA and Hashimoto2015-RA.

global soil respiration was 67 to 108 Pg C yr$^{-1}$ according to different estimates; thus global RA varied from 25 to 51 Pg C yr$^{-1}$. The developed RF-RA dataset fell into this range. Similarly, RA increased by $0.025 \pm 0.006$ Pg C yr$^{-2}$ during 1980–2012. Such an increase may be related to the increasing photosynthesis due to global warming and $CO_2$ fertilization effects, which could increase carbon availability in plant-derived substrate inputs into the soil (e.g., root exudates and biomass) for both root metabolisms (Piñeiro et al., 2017; Zhou et al., 2016). This annual increase accounted for about 25 % of the global soil respiration increase (0.09 and 0.1 Pg C yr$^{-2}$; Bond-Lamberty and Thomson, 2010; Hashimoto et al., 2015), suggesting that about one-quarter of the total soil respiration increment due to climate change came from RA.

With a 1 °C increase in global mean temperature, RA will increase by $0.85 \pm 0.13$ Pg C yr$^{-2}$ and $0.17 \pm 0.03$ Pg C yr$^{-2}$ for a 10 mm increase in precipitation, indicating that carbon fluxes from RA might positively feedback to future climate change, which was typically characterized by increasing temperature and changes in precipitation (IPCC, 2013). However, the RA increment varied with climate zones and ecosystem types (Figs. S2 and S3), which was similar to previous findings in which total soil respiration or RA varied with climate zones or ecosystem types (Ballantyne et al., 2017; Jian et al., 2018a). These differences may be related to regional heterogeneity and the plant functional trait. For example, regional temperature significantly differed from global averages (Huang et al., 2012), with much faster change in high-latitude regions (Hartmann et al., 2014), and semi-arid climates dominated the trend and variability in global land $CO_2$ sink (Ahlström et al., 2015). Therefore, the regionally uneven responses of RA to climatic variables highlight the urgent need to account for regional heterogeneity when studying the effects of climate change on ecosystem carbon dynamics in future.

RF-RA also has important indications of carbon allocation from photosynthesis. The immediate carbon substrates for RA were primarily derived from recent photosynthesis (Högberg et al., 2001; Subke et al., 2011). Strong correlation between photosynthesis and RA demonstrated the evidence for their close coupling relationships (Chen et al., 2014; Kuzyakov and Gavrichkova, 2010). Globally, GPP was about 125 Pg C yr$^{-1}$ during last few decades (Bodesheim et al., 2018; Zhang et al., 2017). Thus, roots respired more than one-third of carbon from GPP, suggesting that except the carbon used for constructing belowground tissues, a large proportion of carbon will be returned back to the atmosphere respired by roots. However, it should be noted that through root respiration, soil nutrients for vegetation growth will be required, which may affect the RA flux.

## 5.2 Dominant factors

Spatially, the dominant driving factors for RA varied greatly. Temperature and shortwave radiation were the main driving factors for high-latitude areas above 50° N (Fig. 9a). This result was not surprising because RA was positively correlated with temperature or photosynthesis (indirectly reflecting the solar radiation) (Chen et al., 2014; Tang et al., 2016), and high-latitude regions was always limited by temperature or energy, leading to low RA as well (Fig. 3a).

Globally, precipitation was the most important factor, covering about 56 % of land (Figs. 9a and 10). Precipitation was always considered to be a proxy for soil water content (Hursh et al., 2017; Yao et al., 2018), and such wide dominance of precipitation on RA was related to the mechanisms of soil water availability driving RA. First, soil water exists in the form of ice when temperature is below zero. In this case, plant and soil microbes cannot directly use it for growth or respiration. This could be observed in some boreal areas where precipitation was the dominant factor of RA (Fig. 9a). Second, soil water content that is too high or too low (e.g., flooding and drought) could limit the mobility of substrates and carbon input to below the ground, which could affect RA. Yan et al. (2014) found that soil respiration decreased once soil water content was below a lower (14.8 %) or above an upper (26.2 %) threshold in a poplar plantation. Similarly, Gomez-Casanovas et al. (2012) also found that RA decreased when soil water content was above 30 %. These results seemed to support our findings. Third, the relationship between soil water content and RA or total soil respiration is more complex than the relationship between temperature and soil respiration. Numerous formulas, such as linear (Tang et al., 2016), polynomial (Moyano et al., 2012), logarithmic (Schaefer et al., 2009) or quadratic (Hursh et al., 2017) models, have been widely applied to describe the relationship between soil water content and soil respiration. The multifarious relationships between soil water content and RA may occur because soil water content affects RA in multiple ways. Meanwhile, seasonal variability in precipitation and soil water content is often correlated with temperature (Feng and Liu, 2015), making the relationship between soil water content and RA more complex.

Similarly, the dominance of precipitation in Hashimoto was also widely observed (Fig. 8), dominating 77 % of land (Fig. 10). Although this percentage was 17 % higher than RF-RA, both results demonstrated that the global RA in the majority of land was dominated by precipitation. However, it is noticeable that the dominant environmental factor controlling spatial carbon fluxes gradient differs among different years (Reichstein et al., 2007), e.g., for extreme climates and climatic disturbance.

## 5.3 Comparison with Hashimoto2015-RA

Globally, total RF-RA was slightly higher than Hashimoto2015-RA; however, great divergence was observed both spatially and temporally (Fig. 6), particularly in tropical regions, where RF-RA was much lower than Hashimoto2015-RA (Fig. 3). These differences could be attributed to several reasons. First, two RA datasets had different land cover areas, especially in desert areas in North Africa, where very sparse or no vegetation existed. If RF-RA was masked by Hashimoto2015-RA, global RA was $39.6 \pm 0.4\,\mathrm{Pg\,C\,yr^{-1}}$, which was pretty close to Hashimoto2015-RA (Fig. S8). Second, different predictors and algorithms were applied for RF-RA and Hashimoto2015-RA prediction. Besides temperature and precipitation, RA was also affected by soil nutrients, carbon substrate supply, belowground carbon allocation, site disturbance and other variables (Chen et al., 2014; Hashimoto et al., 2015; Tang et al., 2016; Zhou et al., 2016). Hashimoto2015-RA was calculated from the difference between total soil respiration and heterotrophic respiration, which were predicted by a simple climate-driven model using temperature and precipitation only (Hashimoto et al., 2015). Thus, Hashimoto2015-RA could not reflect its soil nutrient and other environmental constraints. To overcome such limitations, besides temperature and precipitation, we included soil water content, soil nitrogen and soil organic carbon as proxies for environmental and nutrient constraints of RA and considered the interactions among these variables using RF, achieving a model efficiency of 0.52 for RA prediction (Fig. S1), which was higher than that for Hashimoto soil respiration, with a model efficiency of 0.32 (Hashimoto et al., 2015). The simple climate model for Hashimoto soil respiration has advantages and limitations (Hashimoto et al., 2015). Third, the empirical model (the relationship between total soil respiration and heterotrophic respiration) from which Hashimoto2015-RA is derived originated from forest ecosystems (Bond-Lamberty et al., 2004; Hashimoto et al., 2015), which may bring uncertainties to other ecosystems. For example, the difference between RF-RA and Hashimoto2015-RA varied by up to $350\,\mathrm{g\,C\,yr^{-1}}$ in southern and northern Amazon areas and in Madagascar, where the savannas were widely distributed (Fig. 4); thus Hashimoto2015-RA might not capture the spatial and temporal pattern of RA for non-forest ecosystems. Including more environmental variables and improving the algorithm could be a great advantage to reduce the uncertainty in modeling RA.

## 5.4 Advantages, limitations and uncertainties

Generally, the developed RF-RA dataset had four main advantages in estimating global RA: first, the RF-RA dataset, to our knowledge, was the first attempt to model RA using a large number of empirical field observations, and the spatial and temporal patterns of RA were investigated globally. In contrast, most previous studies mainly focused on global soil respiration, which was not partitioned into RA and heterotrophic respiration globally (Hursh et al., 2017; Jian et al., 2018b; Zhao et al., 2017). Second, we used an up-to-date field observational dataset developed from the SRDB up to November 2018 (Bond-Lamberty and Thomson, 2018) and updated it by including 68 observations from Chinese peer-reviewed literature. This new updated dataset included a total of 449 field observations (Fig. 1). These observations had a wide coverage range of global terrestrial ecosystems and represented all major biomes and climate zones. Third, the global terrestrial ecosystems were separated into eight

biomes, including boreal forest, cropland, grassland, savannas, shrubland, temperate forest, tropical forest and wetland. The total RA and its inter-annual variability were evaluated for each of the eight biomes (Figs. S3 and S4). Besides, total RA and its inter-annual variability were also assessed for three climate zones – boreal, temperate and tropical zones (Figs. S2 and S5) – according to the Köppen–Geiger climate classification system (Peel et al., 2007). Different temporal change trends across biomes and climate zones also further indicated uneven responses of RA to climate change across the globe. Fourth, we used a RF algorithm to model and map global RA with the linkage of climate, soil and other environmental predictors. The results showed that RF could capture the spatial and temporal variability well in RA (Fig. S1). Compared to linear regressions for soil respiration prediction (because there was no global RA prediction before this study) with a model efficiency of less than 35 % (Bond-Lamberty and Thomson, 2010; Hashimoto et al., 2015; Hursh et al., 2017), the RF algorithm achieved a much higher model efficiency, at 52 %, which indeed improved the RA modeling and reduced the uncertainties.

Although data-derived global RA could serve as a benchmark for global-carbon-cycle modeling, and the RF-RA filled the data gaps of global RA, limitations and uncertainties still remained in a few aspects. First, although we conducted a data quality control to develop the RF-RA dataset, a lack of a reliable approach for separating RA and heterotrophic respiration may lead to an important uncertainty of RA estimates. There are several approaches, e.g., trenching, stable or radioactive isotopes, and gridding, to partition soil respiration (Bond-Lamberty et al., 2004; Högberg et al., 2001; Hanson et al., 2000); however, each of these approaches has its own limitations. For example, trenching has been widely applied in partitioning RA and heterotrophic respiration due to its easy operation and low cost. On the other hand, heterotrophic respiration may be increased due to the termination of water uptake by roots and the decomposition of remaining dead roots in trenching plots (Hanson et al., 2000; Tang et al., 2016). Commonly, RA was calculated from the difference between total soil respiration and heterotrophic respiration; thus the trenching approach might lead to an underestimation of RA. In our dataset, a total of 254 RA observations were estimated by the trenching approach, while the rest RA observations were estimated by other separation approaches, e.g., isotope, radiocarbon and mass balance. Thus, inconsistent separation approaches could also be another source of uncertainty of RA values.

Second, due to the limited observations of RA at a daily or monthly scale, the RF-RA dataset was produced at an annual scale. Although there was no direct study to compare the difference of RA upscaling from daily or monthly and annual scale, substantial differences of soil respiration upscaling from daily or monthly and annual scales (Jian et al., 2018b) indirectly illustrated the potential difference of RA upscaling from different timescales.

Third, the effects of rising atmospheric $CO_2$ on root growth were not explicitly represented when developing the RF-RA dataset, although $CO_2$ fertilization effects could partly be represented in the increased temperatures. While the magnitude of $CO_2$ fertilization effects on photosynthesis is still uncertain (Gray et al., 2016), RF or other machine-learning approaches are encouraged for quantifying the uncertainties due to $CO_2$ fertilization.

Fourth, we did not consider the effects of human activities and historical changes in biomes on RA. However, important changes may occur in tropical forest, grassland and cropland during last several decades due to human activities (Hansen et al., 2013; Klein Goldewijk et al., 2011). Thus, changes in biomes should be included in future global RA and carbon cycling modeling. However, the lack of such data is the main constrain of detecting the effects of biome change on RA.

Finally, uneven coverage of observations in the updated dataset would be another source of uncertainties. Although our dataset had a wide range of land cover, the observational sites mainly distributed in China, Europe and North America and were dominated by forest. There was a great lack of observations in areas such as Africa, Australia and Russia and biomes such as tropical forest, shrubland, wetland and cropland. However, our dataset covered all major ecosystem types and climate zones across the globe. RA observations caused bias of RF model towards the regions with more observations. Therefore, including more observations in these areas and biomes without observations should largely increase our capability to assess the spatial and temporal patterns of global RA and contribute to improving the global-carbon-cycle modeling to future climate change.

## 6 Data availability

The developed RF-RA dataset is freely downloadable from https://doi.org/10.6084/m9.figshare.7636193 (Tang et al., 2019), called 'Respiration_autotrophic_belowgroud_ glob_ 1980_2012_yr_half_dgree_TangX.nc", which is a globally gridded RA dataset from 1980 to 2012 with a spatial resolution of 0.5° at an annual scale (expressed in $g\,C\,m^{-2}\,yr^{-1}$; grams of carbon per square meter per year). The RA dataset is provided in NetCDF format (Network Common Data Form).

## 7 Conclusions

Although data-derived RA may serve as a benchmark for ecosystem models, no such study has assessed the global variability in RA with a large number of empirical observations that can help bridge the knowledge gap between local, regional and global scales. The RF-RA dataset filled this knowledge gap by linking field observations and globally gridded environmental variables using an RF algorithm, providing a global RF-RA dataset at a spatial resolution of

$0.5° \times 0.5°$ (longitude $\times$ latitude) at an annual scale from 1980 to 2012. Currently, robust findings include the following.

1. Annual mean RA was $43.8 \pm 0.4\,\mathrm{Pg\,C\,yr^{-1}}$, with a temporally increasing trend of $0.025 \pm 0.006\,\mathrm{Pg\,C\,yr^{-2}}$ over 1980–2012, indicating an increasing carbon return from the roots to the atmosphere.

2. Unevenly temporal and spatial variabilities in varying climate zones and biomes indicated their uneven responses to future climate change, challenging the perspective that the parameters of global carbon stimulation are independent of climate zones and biomes.

3. Precipitation dominated RA for most of the land globally.

4. The RF-RA dataset has great potential to serve as a benchmark for future data–model comparisons to understand the mechanisms of belowground vegetation carbon allocation and its dynamics. However, further improvements in modeling algorithms, including more observations in areas without field measurements, should overcome shortcomings from reduced data availability and the mismatch in spatial resolution between covariates and in situ RA.

**Supplement.** The supplement related to this article is available online at: https://doi.org/10.5194/essd-11-1-2019-supplement.

**Author contributions.** XT, SF, WZ and SG designed the research and collected the data, XT, WZ and SG contributed to the data processing and analysis. XT, WZ, SG, GC and LS wrote the paper, and all authors contributed to the review of the paper.

**Competing interests.** The authors declare that they have no conflict of interest.

**Acknowledgements.** The authors express their great thanks to the contributors to the Hashimoto soil respiration dataset and the contributors of data from the SRDB, with particular thanks to Ben Bond-Lamberty's continuous efforts to improve the SRDB for years. Many thanks to Shoji Hashimoto for their valuable comments that improved the paper. Great thanks to two anonymous reviewers and the editor – Birgit Heim – for their constructive suggestion and comments for improving the paper.

**Financial support.** This study was supported by the National Natural Science Foundation of China (31800365 and 41671432); Fundamental Research Funds of International Centre for Bamboo and Rattan (1632017003 and 1632018003); National Key Research and Development Project (2017YFC1501002

and 2018YFC1504702); Major Scientific and Technological Support Research Subject for the Prevention and Control of Ecological Geological Disasters in "8.8" Jiuzhaigou Earthquake Stricken Area of Department of Natural Resources of Sichuan Province (KJ-2018-20); Innovation funding of Remote Sensing Science and Technology of Chengdu University of Technology (KYTD201501); Starting Funding of Chengdu University of Technology (10912-2018KYQD-06910), Foundation for University Key Teacher of Chengdu University of Technology (10912-2019JX-06910), and Open Funding from the Key Laboratory of Geoscience Spatial Information Technology of Ministry of Land and Resources (Chengdu University of Technology).

**Review statement.** This paper was edited by Birgit Heim and reviewed by two anonymous referees.

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

## Remarks from the typesetter