# Peer review of "Global variability of belowground autotrophic respiration in terrestrial ecosystems"

_Earth System Science Data, 2019_

## Referee Comment (RC1) · Anonymous Referee #1 · 4 Apr 2019

This manuscript deals with estimation of global belowground autotrophic respiration (RA) in terrestrial ecosystems. I have some questions in this study. 1) Authors compared global RA by data-derived with that by Hashimto et al.(2015). However, authors did not refer the data of Hashimoto et al. (2015) in the manuscript. How did authors get the data from Hashimoto? Please explain the difference between Randomforest model and methods of Hashimoto et al. (2015). 2) Authors used PgC a-1 or gC m-2 a-1for the unit of RA, but I guess that a-1 should be yr-1. Please correct all unit in the manuscript and figures. 3) Authors discussed about importance of the dominant environmental factors for estimate spatio-temporal variation in RA. I think that it is important not only environmental factors for plant production but also plant biomass because root respiration would have positive correlation with plant biomass. Why did

authors ignore the global pattern of plant biomass?? ãĂĂFinally, please considering my specific comments and get some English proofreading. In addition, please reconsider carefully about all figures, because I feel that some figures are not important in this manuscript. If authors resolve these questions, I think that this manuscript would be better for global data science.

Specific comments Page 3, line 54, "which is almost 5 times of....": I cannot understand relationship between this sentence and preceding sentence. Page 3, line 56, "Therefore, an accurate estimate of ...": I think that authors did not enough explain the reasons before the sentences. Please add more explanation. Page 8, Figure 2: I cannot the meaning of the figure 2c and 2b. Why did authors indicate the standard deviation of temporal variation in RA?? Page 11, Figure6: what the difference of Fig.6a and Fig.6b? Please add more explanation. And, please make the same value of y axis in both of Fig.6a and Fig.6b. And I think that Fig6c is not needed. Page 12, Line 254 to 227, "All the biomes, except..., respectively": please rewrite these sentences. Grammatical subject is RA, I think. Page 12, Line 259, "a significant increasing trend of...": is this "a significant increasing trend of total RA in temperate zones,...."?? Page 13, Figure9: I cannot understand the importance of this figure. Page 14, Line 290 " For example, temperature was the ...Australia" is that the result of Hashimoto et al. (2015)?

---

## Referee Comment (RC2) · Anonymous Referee #2 · 15 Jun 2019

I have read "Global variability of belowground autotrophic respiration in terrestrial ecosystems". In the manuscript, the authors estimated global belowground autotrophic respiration from 1980-2012, analyzed the temporal trend, and explored the dominant factors for autotrophic variability. Global autotrophic respiration is a big carbon exchange between the atmosphere and terrestrial, but was rarely studies in the past years. Global temporal and spatial variability of autotrophic respiration is clearly a timely and interesting topic. Generally, this manuscript is well organized and easy to follow. The results and conclusions are reasonable. The production (Global belowground autotrophic respiration shared in the figShare) is a contribution to the community and potentially can serve as a benchmark for ecosystem models, it will be useful also make the analysis (include the codes) public available to make the analysis reproducible. But

I think the authors have to better address the limitation, weakness, and uncertainty of this study. In my opinion, some major limitation including:

1) The sample size of RA: there are much less annual RA comparing with annual Rs (less than 10%), even though the authors extended the RA dataset by new papers from China Knowledge Resource Integrated (CNKI) Database, the total samples is only 449. And the majority of the samples are from the forest, samples from wetland and shrubland are extremely lacking (only 5 observations).

2) How can you evaluate the quality of the RA data? Even though the authors conducted quality control on the RA data, but it does not guarantee the reliability of the RA data. We lack reliable methods to separate RA and RH, current ways (e.g., trend, gap, girdling, clip, and isotope) have their own problem. Further, usually RH is measured, and RA was calculated as the difference between RS and RH, which also bring uncertainties. All those issues were not addressed and discussed in the manuscript. If the data reliability cannot be guaranteed, the estimates, trend, and dominant factors should also be questioned.

Despite the above problems, I still think this study tend to address an important topic and may inspire more research in the future.

Specific comments Abstract

Line 22: (srdb v4) but later (line 97) you used (srdb version 4), be consistent.

Line 24: the unit for RA increasing trend should be Pg C a-2? Please see this paper: Ballantyne, A., Smith, W., Anderegg, W., Kauppi, P., Sarmiento, J., Tans, P., Shevliakova, E., et al. (2017). the warming hiatus due to reduced respiration. Nature Climate Change, 7(2), 148. https://doi.org/10.1038/NCLIMATE3204 – 152.

Line 31-32: "the perspective that the parameters of global carbon stimulation independent on climate zones and biomes". But already some studies said that the response of respiration to climate change differs in different regions. Huang, Jian-ping,

[Figure]

Xiao-dan Guan, and Fei Ji. "Enhanced cold-season warming in semi-arid regions." Atmospheric Chemistry and Physics 12.12 (2012): 5391-5398. Jian, Jinshi, et al. "Future global soil respiration rates will swell despite regional decreases in temperature sensitivity caused by rising temperature." Earth's Future 6.11 (2018): 1539-1554. The response of respiration to climate differs in different periods: Ballantyne, A., Smith, W., Anderegg, W., Kauppi, P., Sarmiento, J., Tans, P., Shevliakova, E., et al. (2017). the warming hiatus due to reduced respiration. Nature Climate Change, 7(2), 148. https://doi.org/10.1038/NCLIMATE3204 – 152. Jian, Jinshi, et al. "Future global soil respiration rates will swell despite regional decreases in temperature sensitivity caused by rising temperature." Earth's Future 6.11 (2018): 1539-1554. Introduction

Line 48: It is not accurate to say RA is the second largest source of carbon fluxes from soil because we don't know whether Ra is larger than Rh. And does the (Raich and Schlesinger 192) paper really say that? And in line 309 you said Rh account for 0.54-0.63, means RH > RA.

Line 54: there is a new study summarized global Rs estimates: Bond-Lamberty, Ben. "New techniques and data for understanding the global soil respiration flux." Earth's Future 6.9 (2018): 1176-1180.

Line 62-63: a citation needs to support this statement.

Line 63-64: need a citation.

Line 85: "linear of non-linear models" change to "linear and non-linear models".

Line 86: But in line 94, you said RF model can avoid overfitting. Zhao et al 2017 used ANN models; and Jian et al 2018 also include RF models. So you need to be concise to avoid inconsistent.

Line 95: Zhao et al. 2017 used ANN models, it is not appropriate to cite here.

Line 96: It is better also include the GitHub commit number of SRDB.

[Figure]

Line 105: other environmental factors is too broad, please to be more specific.

Material and methods

A big point in this study is you compared your results with that from Hashimoto (2015), you need to talk about how you get the RA data of Hashimoto (2015). You directly used their data or you reproduced their estimates. If you reproduced, how and whether you used the same climate data as Hashimoto?

Line 110-112: are those papers from CNKI all in Chinese? How many studies and how many more data records you got from that? Please clarify that.

Line 122: Australia, Russia, Africa, and South America.

Line 145: The srdb v4 covered 1960-2017, why your study only covered 1980-2012? Results

Line 224: '-4 – 4' change to '-4 to 4'.

Line 224-225: 'East Russia and tropical and Eastern regions in Africa' change to 'East Russia, tropical, and Eastern regions in Africa'.

Line 264-265: Usually anomaly was the difference between temperature/precipitation of corresponding year to the mean of a period (e.g., 1980-2012 in this study). But this should not change the results, if previous studies calculate anomaly like yours, please provide a citation to support.

Line 270-273: why in temperate zone/savannas/wetland there is no correlation between RA and temperature anomaly? That is interesting, usually, in tropical and subtropical regions, Rs is less correlated with temperature (and should be also true for the temperature anomaly). I think it worth to analyze in more details and try to explain the mechanism or maybe just because of the uncertainty.

Line 310-311: See also Lamberty 2018 Earth's Future paper. "New techniques and data for understanding the global soil respiration flux." Earth's Future 6.9 (2018): 1176-

1180.

Discussion

Dominant factors: all you talked were about driving factors of RA spatial variability, right? Did you also analyze the dominant factors of temporal variability? Limitation and uncertainty: see my previous overall comment. In addition, Jian et al. "Constraining estimates of global soil respiration by quantifying sources of variability." Global change biology 24.9 (2018): 4143-4159 talked about uncertainty related to time-scaling and Rs upscaling. How about RA upscaling and timescale?

Author contributions

Line 445: 'to the review the manuscript' change to 'to review the manuscript'.

---

## Author Comment (AC1) · 17 Jul 2019

Dear editor and reviewer,

Thank you very much for your great efforts, comments and suggestion! According to your comments and suggestion, we revised the manuscript carefully and thoroughly. Please see, below, our point-to-point response.

Please do not hesitate to let us know if you have additional questions and/or comments.

Sincerely,

Xiaolu Tang, Wenjie Zhang and Sicong Gao, on behalf of all co-authors.

Response to reviewer #1

This manuscript deals with estimation of global belowground autotrophic respiration (RA) in terrestrial ecosystems. I have some questions in this study. 1) Authors compared global RA by data-derived with that by Hashimto et al.(2015). However, authors did not refer the data of Hashimoto et al. (2015) in the manuscript. How did authors get the data from Hashimoto? Please explain the difference between Random forest model and methods of Hashimoto et al. (2015).

Response: we apologize for the unclear statement of Hashimoto RH.

Hashimto RA was publicly available at:

http://cse.ffpri.affrc.go.jp/shojih/data/index.html, therefore, we obtained the annual RA product for our study. Such information was added in text:

"In order to compare with the solely global RA product generated by Hashimoto et al. (2015), which was estimated by a climate-driven model using temperature and precipitation only and obtained from the public available dataset (http://cse.ffpri.affrc.go.jp/shojih/data/index.html)".

Hashimto et al. (2015) proposed a global RA based on the difference of heterotrophic respiration and total soil respiration, and total soil respiration was predicted by a climate-driven model using temperature and precipitation only and global soil respiration dataset. Therefore, Hashimoto RA did not consider other environmental control, such as soil carbon, on RA (Hashimoto et al., 2015).

To fill such knowledge gap, we applied a Random Forest algorithm to model global RA with field observations and 11 environmental variables in terms of different aspects of environmental controls on RA, and we obtained a much higher model efficiency (52%) compared to Hashimoto RA (32%). Furthermore, Random Forest algorithm have great potentials to address the non-linear correlation between RA and environmental variables, and remove auto-correlations among environmental variables.

2) Authors used PgC a-1 or gC m-2 a-1for the unit of RA, but I guess that a-1 should be yr-1. Please correct all unit in the manuscript and figures.

Response: yes, it means Pg C per year. Corrected to "Pg C yr$^{-1}$" or "g C m$^{-2}$ yr$^{-1}$" throughout the manuscript and figures!

3) Authors discussed about importance of the dominant environmental factors for estimate spatio-temporal variation in RA. I think that it is important not only environmental factors for plant production but also plant biomass because root respiration would have positive correlation with plant biomass. Why did authors ignore the global pattern of plant biomass??

Response: thank you for the good comments. We agree with you that plant biomass, particularly root biomass, would have positive correlations with RA. However, selecting variables is constrained by the fact that a variable must be available at all sites and at the corresponding global product simultaneously. For instance, if a variable is measured accurately at sites, but with large uncertainties in the corresponding global product, it may be advantageous to exclude this variable from the analysis (Jung et al., 2011).

Although we tried to include global plant or root biomass as a driving variable, we

found such product was only available for a single year, or mean values of several years (Huang et al., 2017), or forests (Hengeveld et al., 2015), and there was a lack of time-series global biomass product covering all land covers. Given the fact that plant biomass was highly dynamic due to annual accumulation, using a global biomass for a given year or particularly ecosystem type to represent the biomass dynamics covering all terrestrial ecosystems would cause a great uncertainty to RA estimation. Therefore, the lack of global biomass product constrained the use of plant biomass as a driving variable for RA in this study. Instead, we used MODIS land cover as one of driving variables, which could indirectly reflect the biotic or biomass control on RA to some extent.

Finally, please considering my specific comments and get some English proofreading. In addition, please reconsider carefully about all figures, because I feel that some figures are not important in this manuscript. If authors resolve these questions, I think that this manuscript would be better for global data science.

Response: we answered each of your specific comment carefully, and we improved the English.

As you suggested, see specific comments below, Figure 6c was not important and removed.

Specific comments

Page 3, line 54, "which is almost 5 times of: : :.": I cannot understand relationship between this sentence and preceding sentence. Page 3, line 56, "Therefore, an accurate estimate of : : :": I think that authors did not enough explain the reasons before the sentences. Please add more explanation.

Response: Since the two comments link with each other, we answer the two comments together.

We apologize for the unclear statement. We revised and added more explanation for it

as follows:

"RA could amount roughly up to 54 Pg C yr$^{-1}$ (1 Pg = $10^{15}$ g, calculating RA as an approximate ratio of 0.5 of soil respiration, more details in Hanson et al., 2000) according to different estimates of global soil respiration (Bond-Lamberty, 2018), which is almost 5 times of the carbon release from human activities (Le Quéré et al., 2018). However, the contribution of RA to soil respiration varied greatly from 10% to 90% across biomes, climate zones and among years (Hanson et al., 2000), leading to the strong spatial and temporal variability in RA. Thus, whether RA varies with ecosystem types or climate zones remains an open question at the global scale (Ballantyne et al., 2017). Consequently, an accurate estimate of RA and its spatial-temporal dynamics are critical to understand the response of terrestrial ecosystems to global carbon cycling and climate change."

Page 8, Figure 2: I cannot understand the meaning of the figure 2c and 2b. Why did authors indicate the standard deviation of temporal variation in RA??

Response: Fig. 2b is the mean value of Hashimoto RA over 1980-2012, while Fig. 2c represents the standard deviation of predicted RA in this study. The figure caption was revised as:

"Spatial patterns of annual mean and standard deviation of belowground autotrophic respiration (RA) from 1980 to 2012 for this study (a, c) and Hashimoto RA (b, d), respectively"

Due to the inter-annual variability of environmental controls on RA, RA varied annually. Although Fig. 6 describes the annual variability of total RA, the spatial pattern of annual variability of RA is lacking. To characterize the spatial pattern of annual variability of RA, the standard deviation of RA from 1980-2012 was employed. Such analysis was also conducted in other studies, e.g. Yao et al. (2018). Therefore, we used standard deviation to represent the temporal pattern of RA.

Page 11, Figure6: what the difference of Fig.6a and Fig.6b? Please add more explanation. And, please make the same value of yaxis in both of Fig.6a and Fig.6b.

And I think that Fig6c is not needed.

Response: Fig. 6a represents the annual variability of predicted RA in this study, while Fig. 6b represents the annual variability of Hashimoto RA. The same value of yaxis from 39 – 45 Pg C yr$^{-1}$ was applied.

Fig. 6c was not important and removed.

We corrected the figure description more clearly:

"Figure 6 Annual variability of belowground autotrophic respiration (RA) for this study (a) and Hashimoto RA (b) from 1980 to 2012. The grey area represents 95% confidence interval."

Page 12, Line 254 to 227, "All the biomes, except: : :, respectively": please rewrite these sentences. Grammatical subject is RA, I think.

Response: we rewrite these sentences:

"RA showed a significantly increasing trend during 1980-2012 ($p_s < 0.01$) in most of the biomes, except temperate forest, savannas and wetland. RA in tropical forests, boreal forests and cropland increased by 0.0076±0.0015, 0.0047±0.0016, 0.0036±0.0014 Pg C yr$^{-2}$, respectively."

Page 12, Line 259, "a significant increasing trend of: : :": is this "a significant increasing trend of total RA in temperate zones,: : :."??

Response: thank you for your careful revision. Yes, we mean "a significant increasing trend of total RA in temperate zones….". We revised the text:

"there were significant increasing trends of total RA in temperate zones, temperate forest, savannas and wetland of Hashimoto RA, which were not observed in data-derived RA".

Page13, Figure 9: I cannot understand the importance of this figure.

Response: We appreciate your question. Figure 9 showed the relative importance of three main environmental drivers – MAT, MAP and SWR, by colors with RGB plot.

Due to different ecosystem types, or plant functional types or climate zones, the dominant factors may vary. As indicated by Fig. S7, 56% of land area was dominated by precipitation, while temperature and shortwave radiation dominated 19% and 25% of global land areas, which indicated an uneven control of environmental factors on RA. Therefore, Figure 9 showed the spatial variability of dominance of MAT, MAP and SWR on RA. It was found that the dominance of precipitation on RA was globally distributed, particularly dry or semi-arid areas, such as Northwest China, Southern Africa, Middle Australia and America, while temperature controlled RA mainly in in tropical Africa, Southern Amazon rainforests, Siberia and partly tundra, and shortwave radiation dominated high latitudinal areas, e.g. Eastern America and middle and Eastern Russian. Such analysis have been widely used in other studies, e.g. gross primary production (Yao et al., 2018), earth greening (Zhu et al., 2016), vegetation productivity (Seddon et al., 2016).

RGB synthesis (Fig. 9) was performed on stretched values of partial correlation coefficients, an effective way to illustrate the spatial distribution of dominant driving factors of RA (Yao et al., 2018), which could increase our understanding the mechanisms and spatial variability of environmental controls on RA at the global scale.

Page 14, Line 290 "For example, temperature was the: : :Australia" is that the result of Hashimoto et al.(2015)?

Response: thank you for your careful revision again. Yes, we mean "Hashimoto RA", and revised in text:

"temperature was the main dominant factor for most area of Australia for Hashimoto RA".

References

Ballantyne, A., Smith, W., Anderegg, W., Kauppi, P., Sarmiento, J., Tans, P., Shevliakova, E., Pan, Y., Poulter, B., Anav, A., Friedlingstein, P., Houghton, R., and Running, S.: Accelerating net terrestrial carbon uptake during the warming hiatus due

to reduced respiration, Nature Clim. Change, 7, 148-152, http://dx.doi.org/10.1038/nclimate3204, 2017.

Bond-Lamberty, B.: New Techniques and Data for Understanding the Global Soil Respiration Flux, Earth's Future, 6, 1176-1180, http://dx.doi.org/10.1029/2018ef000866, 2018.

Hanson, P. J., Edwards, N. T., Garten, C. T., and Andrews, J. A.: Separating root and soil microbial contributions to soil respiration: A review of methods and observations, Biogeochemistry, 48, 115-146, http://dx.doi.org/10.1023/a:1006244819642, 2000.

Hashimoto, S., Carvalhais, N., Ito, A., Migliavacca, M., Nishina, K., and Reichstein, M.: Global spatiotemporal distribution of soil respiration modeled using a global database, Biogeosciences, 12, 4121–4132, http://dx.doi.org/10.5194/bgd-12-4331-2015, 2015.

Hengeveld, G. M., Gunia, K., Didion, M., Zudin, S., Clerkx, A. P. P. M., and Schelhaas, M. J.: Global 1-degree Maps of Forest Area, Carbon Stocks, and Biomass, 1950-2010. ORNL Distributed Active Archive Center, https://doi.org/10.3334/ORNLDAAC/1296, 2015.

Huang, J., Li, Y., Fu, C., Chen, F., Fu, Q., Dai, A., Shinoda, M., Ma, Z., Guo, W., Li, Z., Zhang, L., Liu, Y., Yu, H., He, Y., Xie, Y., Guan, X., Ji, M., Lin, L., Wang, S., Yan, H., and Wang, G.: Dryland climate change: Recent progress and challenges, Rev. Geophys., 55, 719-778, http://dx.doi.org/10.1002/2016rg000550, 2017.

Jung, M., Reichstein, M., Margolis, H. A., Cescatti, A., Richardson, A. D., Arain, M. A., Arneth, A., Bernhofer, C., Bonal, D., Chen, J. Q., Gianelle, D., Gobron, N., Kiely, G., Kutsch, W., Lasslop, G., Law, B. E., Lindroth, A., Merbold, L., Montagnani, L., Moors, E. J., Papale, D., Sottocornola, M., Vaccari, F., and Williams, C.: Global patterns of land-atmosphere fluxes of carbon dioxide, latent heat, and sensible heat derived from eddy covariance, satellite, and meteorological observations, J. Geophys. Res. Biogeosci., 116, G00J07, http://dx.doi.org/10.1029/2010jg001566, 2011.

Le Quéré, C., Andrew, R. M., Friedlingstein, P., Sitch, S., Pongratz, J., Manning, A. C., Korsbakken, J. I., Peters, G. P., Canadell, J. G., Jackson, R. B., Boden, T. A., Tans, P. P., Andrews, O. D., Arora, V. K., Bakker, D. C. E., Barbero, L., Becker, M., Betts, R. A., Bopp, L., Chevallier, F., Chini, L. P., Ciais, P., Cosca, C. E., Cross, J., Currie, K., Gasser, T., Harris, I., Hauck, J., Haverd, V., Houghton, R. A., Hunt, C. W., Hurtt, G., Ilyina, T., Jain, A. K., Kato, E., Kautz, M., Keeling, R. F., Klein Goldewijk, K., Körtzinger, A., Landschützer, P., Lefèvre, N., Lenton, A., Lienert, S., Lima, I., Lombardozzi, D., Metzl, N., Millero, F., Monteiro, P. M. S., Munro, D. R., Nabel, J. E. M. S., Nakaoka, S.-i., Nojiri, Y., Padín, X. A., Peregon, A., Pfeil, B., Pierrot, D., Poulter, B., Rehder, G., Reimer, J., Rödenbeck, C., Schwinger, J., Séférian, R., Skjelvan, I., Stocker, B. D., Tian, H., Tilbrook, B., van der Laan-Luijkx, I. T., van der Werf, G. R., van Heuven, S., Viovy, N., Vuichard, N., Walker, A. P., Watson, A. J., Wiltshire, A. J., Zaehle, S., and Zhu, D.: Global Carbon Budget 2017, Earth Syst. Sci. Data, 10, 405-448, http://dx.doi.org/10.5194/essd-2017-123, 2018.

Seddon, A. W., Macias-Fauria, M., Long, P. R., Benz, D., and Willis, K. J.: Sensitivity of global terrestrial ecosystems to climate variability, Nature, 531, 229-232, http://dx.doi.org/10.1038/nature16986, 2016.

Yao, Y., Wang, X., Li, Y., Wang, T., Shen, M., Du, M., He, H., Li, Y., Luo, W., Ma, M., Ma, Y., Tang, Y., Wang, H., Zhang, X., Zhang, Y., Zhao, L., Zhou, G., and Piao, S.: Spatiotemporal pattern of gross primary productivity and its covariation with climate in China over the last thirty years, Glob. Chang. Biol., 24, 184-196, http://dx.doi.org/10.1111/gcb.13830, 2018.

Zhu, Z., Piao, S., Myneni, R. B., Huang, M., Zeng, Z., Canadell, J. G., Ciais, P., Sitch, S., Friedlingstein, P., Arneth, A., Cao, C., Cheng, L., Kato, E., Koven, C., Li, Y., Lian, X., Liu, Y., Liu, R., Mao, J., Pan, Y., Peng, S., Peñuelas, J., Poulter, B., Pugh, T. A. M., Stocker, B. D., Viovy, N., Wang, X., Wang, Y., Xiao, Z., Yang, H., Zaehle, S., and Zeng, N.: Greening of the Earth and its drivers, Nature Climate Change, 6, 791-795, http://dx.doi.org/10.1038/nclimate3004, 2016.

---

## Author Comment (AC2) · 17 Jul 2019

Dear editor and reviewer,

Thank you very much for your great efforts, comments and suggestion! According to your comments and suggestion, we revised the manuscript carefully and thoroughly. Please see, below, our point-to-point response.

Please do not hesitate to let us know if you have additional questions and/or comments.

Sincerely,

Xiaolu Tang, Wenjie Zhang and Sicong Gao, on behalf of all co-authors.

Response to Reviewer #2

I have read "Global variability of belowground autotrophic respiration in terrestrial ecosystems". In the manuscript, the authors estimated global belowground autotrophic respiration from 1980-2012, analyzed the temporal trend, and explored the dominant factors for autotrophic variability. Global autotrophic respiration is a big carbon exchange between the atmosphere and terrestrial, but was rarely studies in the past years. Global temporal and spatial variability of autotrophic respiration is clearly a timely and interesting topic. Generally, this manuscript is well organized and easy to follow. The results and conclusions are reasonable. The production (Global belowground autotrophic respiration shared in the figShare) is a contribution to the community and potentially can serve as a benchmark for ecosystem models, it will be useful also make the analysis (include the codes) public available to make the analysis reproducible. But I think the authors have to better address the limitation, weakness, and uncertainty of this study. In my opinion, some major limitation including: 1) The sample size of RA: there are much less annual RA comparing with annual Rs (less than 10%), even though the authors extended the RA dataset by new papers from China Knowledge Resource

Integrated (CNKI) Database, the total samples is only 449. And the majority of the samples are from the forest, samples from wetland and shrubland are extremely lacking (only 5 observations).

Response: we also attached the dataset and the R codes to generate the main results to figshare at https://doi.org/10.6084/m9.figshare.7636193.

Based on SRDB v4, including new observations from CNKI, we got a total of 4276 observations for soil respiration, however, there were 697 observations for RA. According to our selecting criteria: e.g. RA measurement lasting for one year; excluding measurements with Alkali absorption and soda lime; no site management, we got a RA dataset of 449 observations. Our dataset are mainly from forests, but a lack of observations in wetland and shrubland, which could be the limitation in this study.

We have discussed the limitation in "**4.4 Advantages, limitations and uncertainties**" section as follows:

"Finally, uneven coverage of observations in the updated database would be another source of uncertainties. Although our dataset had a wide range of land cover, the observational sites mainly distributed in China, Europe and North America and were dominated by forests. There was a great lack of observations in areas, such as Africa, Austria and Russia, and biomes, such as tropical forest, shrubland, wetland and cropland. Consequently, RA observations caused bias of RF model toward the regions with more observations."

2) How can you evaluate the quality of the RA data? Even though the authors conducted quality control on the RA data, but it does not guarantee the reliability of the RA data. We lack reliable methods to separate RA and RH, current ways (e.g., trend, gap, girdling, clip, and isotope) have their own problem. Further, usually RH is measured, and RA was calculated as the difference between RS and RH, which also bring uncertainties. All those issues were not addressed and discussed in the manuscript. If the data reliability cannot be guaranteed, the estimates, trend, and dominant factors should also be questioned. Despite the above problems, I still think this study tend to

address an important topic and may inspire more research in the future.

Response: we evaluate the quality of RA from different aspects to guarantee the reliability of RA: (1) measuring approaches: Alkali absorption and soda lime were not included due to the potential underestimate of respiration rate with the increasing pressure inside chamber (Pumpanen et al., 2004); (2) data quality control by quality flag: Q01 (estimated from figure), Q02 (data from another study), Q03 (data estimated-other), Q04 (potentially useful future data), Q10 (potential problem with data), Q11 (suspected problem with data), Q12 (known problem with data), Q13 (duplicate?), Q14 (inconsistency). Therefore, RA or total soil respiration observations labelled by "Q10", "Q11", "Q12", "Q13" and "Q14" were removed in this study. More details on data quality controls can be found in Bond-Lamberty and Thomson (2010a).

We agree with you that there was a lack reliable method to separate RA and RH, and current ways (e.g., trend, gap, girdling, clip, and isotope) have their own problem.

We have discussed the data quality and limitation of unreliable method to separate RA and RH in "**4.4 Advantages, limitations and uncertainties**" as follows:

"First, although we conducted a data quality control in this study, a lack of reliable approach to separate RA and heterotrophic respiration may lead to an uncertainty of RA values. There are several approaches, e.g. trenching, stable or radioactive isotope, gridding (Bond-Lamberty et al., 2004; Högberg et al., 2001; Hanson et al., 2000), however, each of these approaches has its own limitations. For example, trenching has been widely applied to partition RA and heterotrophic respiration due to easy operation and low cost, on the other hand, heterotrophic respiration may be increased due to the termination of water uptake by roots and the decomposition of remaining dead roots in trenching plots (Hanson et al., 2000; Tang et al., 2016). Commonly, RA was calculated from the difference between total soil respiration and heterotrophic respiration, thus the trenching approach might lead to an underestimation of RA. In our dataset, a total of 254 RA observations were estimated by trenching approach, while the rest RA observations were estimated by other separation approaches, e.g. isotope, radiocarbon,

mass balance. Thus, inconsistent separation approaches could be another source of uncertainty of RA values."

Specific comments Abstract

Line 22: (srdb v4) but later (line 97) you used (srdb version 4), be consistent.

Response: done!

Line 24: the unit for RA increasing trend should be Pg C a-2? Please see this paper: Ballantyne, A., Smith, W., Anderegg, W., Kauppi, P., Sarmiento, J., Tans, P., Shevliakova, E., et al. (2017). the warming hiatus due to reduced respiration. Nature Climate Change, 7(2), 148. https://doi.org/10.1038/NCLIMATE3204 – 152.

Response: thank you for your kind recommendation, and we corrected the increasing unit to Pg a $yr^{-2}$ or g C $m^{-2}$ $yr^{-2}$ throughout the text and figures.

Line 31-32: "the perspective that the parameters of global carbon stimulation independent on climate zones and biomes". But already some studies said that the response of respiration to climate change differs in different regions. Huang, Jian-ping, Xiao-dan Guan, and Fei Ji. "Enhanced cold-season warming in semi-arid regions." Atmospheric Chemistry and Physics 12.12 (2012): 5391-5398. Jian, Jinshi, et al. "Future global soil respiration rates will swell despite regional decreases in temperature sensitivity caused by rising temperature." Earth's Future 6.11 (2018): 1539-1554. The response of respiration to climate differs in different periods: Ballantyne, A., Smith, W., Anderegg, W., Kauppi, P., Sarmiento, J., Tans, P., Shevliakova, E., et al. (2017). the warming hiatus due to reduced respiration. Nature Climate Change, 7(2), 148.

https://doi.org/10.1038/NCLIMATE3204 – 152.

Response: thank you for your kind recommendation. Huang et al. (2012) mainly discussed the uneven changes of temperature, not RA.

Jian et al. (2018) found uneven changes of soil respiration in different areas, and Ballantyne et al. (2017) also proposed that belowground autotrophic respiration may be varied among ecosystem types. These references have been cited to support our

conclusions, and revised in the text as follows:

"However, RA increment varied with climate zones and ecosystem types (Figs. S2 and S3), which was similar to previous findings (Ballantyne et al., 2017; Jian et al., 2018a), who found that total soil respiration or RA varied with climate zones or ecosystem types."

Introduction

Line 48: It is not accurate to say RA is the second largest source of carbon fluxes from soil because we don't know whether Ra is larger than Rh. And does the (Raich and Schlesinger 192) paper really say that? And in line 309 you said Rh account for 0.54-0.63, means RH > RA.

Response: we apologize for the improper statement. We mean soil respiration is the second largest carbon flux. We revise the text:

"RA is one main component of soil respiration (Hanson et al., 2000), and soil respiration represents the second largest source of carbon fluxes from soil to the atmosphere (after gross primary production, GPP) in the global carbon cycle (Raich and Schlesinger, 1992)."

Line 54: there is a new study summarized global Rs estimates: Bond-Lamberty, Ben. "New techniques and data for understanding the global soil respiration flux." Earth's Future 6.9 (2018): 1176-1180.

Response: thank you for your recommendation. We cited the global estimates of soil respiration summarized by Bond-Lamberty (2018) to support our study.

"RA could amount roughly up to 54 Pg C yr$^{-1}$ (1 Pg = $10^{15}$ g, calculating RA as an approximate ratio of 0.5 of soil respiration, more details in Hanson et al., 2000) according to different estimates of global soil respiration (Bond-Lamberty, 2018), which is almost 5 times of the carbon release from human activities (Le Quéré et al., 2018)."

Line 62-63: a citation needs to support this statement.

Response: done! We revised the text as follows:

"the globally spatial and temporal pattern of RA has not been explored and still acts as a "black box" in global carbon cycling (Ballantyne et al., 2017)".

Line 63-64: need a citation.

Response: Revised as follows in the text:

"This "black box" is not well constrained and validated, because most terrestrial ecosystem models and earth system models were commonly calibrated and validated against eddy covariance measurements of net ecosystem carbon exchange (Yang et al., 2013)".

Line 85: "linear of non-linear models" change to "linear and non-linear models".

Response: done!

Line 86: But in line 94, you said RF model can avoid overfitting. Zhao et al 2017 used ANN models; and Jian et al 2018 also include RF models. So you need to be concise to avoid inconsistent.

Response: Zhao et al 2017 was appropriate and removed in L94!

Line 95: Zhao et al. 2017 used ANN models, it is not appropriate to cite here.

Response: Zhao et al. 2017 was removed, while Bodesheim et al., 2018 and Jung et al. 2017 were cited here.

Line 96: It is better also include the GitHub commit number of SRDB.

Response: the doi number was added.

Line 105: other environmental factors is too broad, please to be more specific.

Response: revised! We specified the soil and vegetation factors.

"It will also advance our knowledge of the co-variation of RA with climate, soil and

vegetation factors"

Material and methods

A big point in this study is you compared your results with that from Hashimoto (2015), you need to talk about how you get the RA data of Hashimoto (2015). You directly used their data or you reproduced their estimates. If you reproduced, how and whether you used the same climate data as Hashimoto?

Response: we apologize for the misleading of Hashimoto RA. Hashimoto RA is publicly available at http://cse.ffpri.affrc.go.jp/shojih/data/index.html, therefore, we obtained the annual Hashimoto RA product for our study. Such information was added in text:

"In order to compare with the solely global RA product generated by Hashimoto et al. (2015), which was estimated by a climate-driven model using temperature and precipitation only and obtained from the public available dataset (http://cse.ffpri.affrc.go.jp/shojih/data/index.html)"

Line 110-112: are those papers from CNKI all in Chinese? How many studies and how many more data records you got from that? Please clarify that.

Response: yes, those papers from CNKI are all in Chinese with English abstract. We added 68 more RA observations and revised in the text:

"Finally, this study included a total of 449 field observations (Fig. 1), including 68 observations from CNKI."

Line 122: Australia, Russia, Africa, and South America.

Response: done!

Line 145: The srdb v4 covered 1960-2017, why your study only covered 1980-2012?

Results: from 1960 to 1980, there are only 11 observations, which might bring uncertainties. Our study covered the period until 2012 for easily comparing with Hashimoto RA, which covered the period up to 2012.

Line 224: '-4 – 4' change to '-4 to 4'.

Response: done!

Line 224-225: 'East Russia and tropical and Eastern regions in Africa' change to 'East Russia, tropical, and Eastern regions in Africa'.

Response: done!

Line 264-265: Usually anomaly was the difference between temperature/precipitation of corresponding year to the mean of a period (e.g., 1980-2012 in this study). But this should not change the results, if previous studies calculate anomaly like yours, please provide a citation to support.

Response: thank you for your suggestion, and we followed the suggestion. The anomaly of temperature/precipitation of corresponding year to the mean of 1980-2012, and the results did not change.

Line 270-273: why in temperate zone/savannas/wetland there is no correlation between RA and temperature anomaly? That is interesting, usually, in tropical and subtropical regions, Rs is less correlated with temperature (and should be also true for the temperature anomaly). I think it worth to analyze in more details and try to explain the mechanism or maybe just because of the uncertainty.

Response: the different responses of ecosystem types or climate zones to climatic variables may be related to regional heterogeneity and plant functional trait. For example, regional temperature significantly differed from global averages (Huang et al., 2012), with much faster change in high-latitude regions (Hartmann et al., 2014), and semi-arid dominated the trend and variability of global land $CO_2$ sink (Ahlström et al., 2015). Similar studies were also found in other studies, e.g. total soil respiration or RA (Ballantyne et al., 2017; Jian et al., 2018a). Therefore, the regionally uneven responses of RA to climatic variables were unlikely due to model uncertainty.

These results have been discussed in "**4.1 Global RA**" section, and we revised the text as:

"However, RA increment varied with climate zones and ecosystem types (Figs. S2 and S3), which was similar to previous findings (Ballantyne et al., 2017; Jian et al., 2018a), who found that total soil respiration or RA varied with climate zones or ecosystem types. These differences may be related to regional heterogeneity and plant functional trait. For example, regional temperature significantly differed from global averages (Huang et al., 2012), with much faster change in high-latitude regions (Hartmann et al., 2014), and semi-arid dominated the trend and variability of global land $CO_2$ sink (Ahlström et al., 2015). Therefore, the regionally uneven responses of RA to climatic variables highlights the urgent need to account for regional heterogeneity when studying the effects of climate change on ecosystem carbon dynamics in future."

Line 310-311: See also Lamberty 2018 Earth's Future paper. "New techniques and data for understanding the global soil respiration flux." Earth's Future 6.9 (2018): 1176-1180.

Response: thank you for the recommendation, and we cited the global soil respiration estimates from Bond-Lamberty (2018):

"Bond-Lamberty et al. (2018) proposed that the global average proportion of heterotrophic respiration ranged from 0.54 to 0.63 over 1990-2014 and global total soil respiration was 67 to 108 Pg C $yr^{-1}$ using different approaches and datasets Bond-Lamberty (2018); (Bond-Lamberty and Thomson, 2010b; Hashimoto et al., 2015; Hursh et al., 2017; Jian et al., 2018b) , thus global RA varied from 25 to 51 Pg C $yr^{-1}$."

Discussion

Dominant factors: all you talked were about driving factors of RA spatial variability, right? Did you also analyze the dominant factors of temporal variability? Limitation and uncertainty: see my previous overall comment. In addition, Jian et al. "Constraining estimates of global soil respiration by quantifying sources of variability." Global change biology 24.9 (2018): 4143-4159 talked about uncertainty related to time-scaling and Rs upscaling. How about RA upscaling and timescale?

Response: we analyzed the dominate factors at both spatial and temporal patterns. We

used partial correlation analysis based on a timescale from 1980 to 2012 for each grid cell (see methodology section 2.5), and the correlation coefficient was applied to derive the dominant factor map (Fig. 9). However, we did not analyze the dominant factors for each given year.

We additionally discussed the potential variability of RA using different time scale variables in "**4.4 Advantages, limitations and uncertainties**".

[revised manuscript text omitted]

---

## Author Comment (AC3) · 28 Aug 2019

Response to Reviewer #1 Dear editor and reviewer,

Thank you very much for your great efforts, comments and suggestion again! Based on the first version of "the response to reviewer", the editor evaluated our manuscript carefully and proposed some good suggestion and comments. Combining editor and reviewers' suggestion and comment, we revised our manuscript carefully and thoroughly. Therefore, I am kindly to remind you that it would be much more efficient to work on the updated "the response to reviewer" and discard the old version that uploaded on July 17, 2019. Thank you for your understanding.

Suggestion and comments from referees or editor are marked in Black. Responses

to referee and editor's comments are labelled in blue. Cited changes made in the manuscript are marked in red.

Please do not hesitate to let us know if you have further questions and/or comments.

Sincerely, Xiaolu Tang, Wenjie Zhang and Sicong Gao, on behalf of all co-authors.

Please also note the supplement to this comment:
https://www.earth-syst-sci-data-discuss.net/essd-2019-18/essd-2019-18-AC3-supplement.pdf

**Supplement:**

**Response to Reviewer #1**

Dear editor and reviewer,

Thank you very much for your great efforts, comments and suggestion again! Based on the first version of "the response to reviewer", the editor evaluated our manuscript carefully and proposed some good suggestion and comments. Combining editor and reviewers' suggestion and comment, we revised our manuscript carefully and thoroughly.

Therefore, I am kindly to remind you that it would be much more efficient to work on **the updated** "the response to reviewer" and **discard the old version that uploaded on July 17, 2019.** Thank you for your understanding.

**Suggestion and comments from referees or editor** are marked in **Black**.

**Responses to referee and editor's comments** are labelled in **blue**.

**Cited changes made in the manuscript** are marked in **red**.

Please do not hesitate to let us know if you have further questions and/or comments.

Sincerely,

Xiaolu Tang, Wenjie Zhang and Sicong Gao, on behalf of all co-authors.

Response to reviewer #1

This manuscript deals with estimation of global belowground autotrophic respiration (RA) in terrestrial ecosystems. I have some questions in this study. 1) Authors compared global RA by data-derived with that by Hashimto et al.(2015). However, authors did not refer the data of Hashimoto et al. (2015) in the manuscript. How did authors get the data from Hashimoto? Please explain the difference between Random forest model and methods of Hashimoto et al. (2015).

Response: we apologize for the unclear statement of Hashimoto RH. To avoid that the readers of this manuscript need to read Hashimoto et al. 2015 to understand the RA dataset, we first added a short summary about the details of input data, the method used and how soil respiration and RA derived. Then we provided details on how to get Hashimoto2015-RA (renamed as suggested by editor to fit ESSD better).

Detailed in text as:

"Hashimoto developed a climate-driven model by updating Raich's model, which stimulated soil respiration as a function of temperature and water (precipitation) at a monthly time step (Hashimoto et al., 2015; Raich et al., 2002). Therefore, to get a global estimate to soil respiration at a monthly scale, the globally gridded air temperature and precipitation with a spatial resolution of 0.5º were derived from University of East Anglia CRU 3.21 (Harris et al., 2014), and 1638 field observations were taken from SRDB (v3) for model parameterization (Hashimoto et al., 2015). Monthly soil respiration was summed to a yearly scale. Furthermore, annual soil respiration was divided into autotrophic and heterotrophic respiration using a global relationship between soil respiration and heterotrophic respiration derived from a meta-analysis (Bond-Lamberty et al., 2004). This global relationship can be expressed by:

$$\ln(RH) = 1.22 + 1.73 \times \ln(RS) \qquad (1)$$

Where RH means annual heterotrophic respiration, and RS stands for annual soil respiration, expressed by g C m$^{-2}$ yr$^{-1}$.

Therefore, global Hashimoto2015-RA was derived by the difference between soil respiration and heterotrophic respiration. The monthly or annual Hashimoto2015-RA dataset can be freely accessed from (http://cse.ffpri.affrc.go.jp/shojih/data/index.html, Hashimoto et al., 2015)."

Therefore, the main differences between Random Forest based RA (RF-RA) and Hashimoto20105-RA included: (1) variables used to develop models. To develop RF-RA, we applied 11 variables, including temperature, precipitation, soil nitrogen, soil carbon and other environmental variables, while on

temperature and precipitation were applied in develop Hashimoto2015-RA; (2) modelling approaches. We used Random Forest algorithm to model RA, while a simple climate-driven model by temperature and precipitation was applied to develop Hashimoto2015-RA. Consequently, we obtained a much higher model efficiency (52%) compared to Hashimoto2015-RA (32%). Furthermore, Random Forest algorithm have great potentials to address the non-linear correlation between RA and environmental variables, and remove auto-correlations among environmental variables.

2) Authors used PgC a-1 or gC m-2 a-1for the unit of RA, but I guess that a-1 should be yr-1. Please correct all unit in the manuscript and figures.

Response: yes, it means Pg C per year. Corrected to "Pg C yr$^{-1}$" or "g C m$^{-2}$ yr$^{-1}$" throughout the manuscript and figures!

3) Authors discussed about importance of the dominant environmental factors for estimate spatio-temporal variation in RA. I think that it is important not only environmental factors for plant production but also plant biomass because root respiration would have positive correlation with plant biomass. Why did authors ignore the global pattern of plant biomass??

Response: thank you for the good comments. We agree with you that plant biomass, particularly root biomass, would have positive correlations with RA. However, selecting variables is constrained by the fact that a variable must be available at all sites and at the corresponding global product simultaneously. For instance, if a variable is measured accurately at sites, but with large uncertainties in the corresponding global product, it may be advantageous to exclude this variable from the analysis (Jung et al., 2011).

Although we tried to include global plant or root biomass as a driving variable, we found such product was only available for a single year, or mean values of several years (Huang et al., 2017), or forests (Hengeveld et al., 2015), and there was a lack of time-series global biomass product covering all land covers. Given the fact that plant biomass was highly dynamic due to annual accumulation, using a global biomass for a given year or particularly ecosystem type to represent the biomass dynamics covering all terrestrial ecosystems would cause a great uncertainty to RA estimation. Therefore, the lack of global biomass product constrained the use of plant biomass as a driving variable for RA in this study. Instead, we used MODIS land cover as one of driving variables, which could indirectly reflect the biotic or biomass control on RA to some extent.

Finally, please considering my specific comments and get some English proofreading. In addition, please

reconsider carefully about all figures, because I feel that some figures are not important in this manuscript. If authors resolve these questions, I think that this manuscript would be better for global data science.

Response: we answered each of your specific comment carefully, and we improved the English carefully.

As you suggested, see specific comments below, Figure 6c was not important and removed.

Specific comments

Page 3, line 54, "which is almost 5 times of: : :.": I cannot understand relationship between this sentence and preceding sentence. Page 3, line 56, "Therefore, an accurate estimate of : : :": I think that authors did not enough explain the reasons before the sentences. Please add more explanation.

Response: Since the two comments link with each other, we answer the two comments together.

We apologize for the unclear statement. We revised and added more explanation for it as follows:

"Globally, RA could amount roughly up to 54 Pg C yr$^{-1}$ (1 Pg = $10^{15}$ g, calculating RA as an approximate ratio of 0.5 of soil respiration, more details in Hanson et al., 2000) according to different estimates of global soil respiration (Bond-Lamberty, 2018), which is almost 5 times of the carbon release from human activities (Le Quéré et al., 2018). However, the contribution of RA to soil respiration varied greatly from 10% to 90% across biomes, climate zones and among years (Hanson et al., 2000), leading to the strong spatial and temporal variability in RA. Thus, whether RA varies with ecosystem types or climate zones remains an open question at the global scale (Ballantyne et al., 2017). Consequently, an accurate estimate of RA and its spatial-temporal dynamics are critical to understand the response of terrestrial ecosystems to climate change."

Page 8, Figure 2: I cannot understand the meaning of the figure 2c and 2b. Why did authors indicate the standard deviation of temporal variation in RA??

Response: Fig. 2b is the mean value of Hashimoto RA over 1980-2012, while Fig. 2c represents the standard deviation of predicted RA in this study. The figure caption was revised combining the other and editor suggestion and comments:

"**Figure 2** Spatial patterns of annual mean and standard deviation for RF-RA (a, c) and Hashimoto2015-RA (b, d) from 1980 to 2012, respectively. The standard deviation was applied to characterize the inter-annual variability following Yao et al. (2018)."

Due to the inter-annual variability of environmental controls on RA, RA varied annually. Although Fig. 6 describes the annual variability of total RA, the spatial pattern of annual variability of RA is lacking. To characterize the spatial pattern of annual variability of RA, the standard deviation of RA from 1980-2012 was employed. Such analysis was also conducted in other studies, e.g. Yao et al. (2018). Therefore, we used standard deviation to represent the temporal pattern of RA.

Page 11, Figure6: what the difference of Fig.6a and Fig.6b? Please add more explanation. And, please make the same value of yaxis in both of Fig.6a and Fig.6b. And I think that Fig6c is not needed.

Response: Fig. 6a represents the annual variability of predicted RA in this study, while Fig. 6b represents the annual variability of Hashimoto RA. The same value of yaxis from $39 - 45$ Pg C $yr^{-1}$ was applied.

Fig. 6c was not important and removed.

We corrected the figure description more clearly:

"Annual variability of RF-RA (a) and Hashimoto2015-RA (b) from 1980 to 2012. The grey area represents 95% confidence interval."

Page 12, Line 254 to 227, "All the biomes, except: : :, respectively": please rewrite these sentences. Grammatical subject is RA, I think.

Response: we rewrite these sentences:

"RA showed a significantly increasing trend during 1980-2012 ($p_s < 0.01$) in majority biomes, except temperate forests, savannas and wetland. RA in tropical forests, boreal forests and cropland increased by $0.0076\pm0.0015$, $0.0047\pm0.0016$, $0.0036\pm0.0014$ Pg C $yr^{-2}$, respectively."

Page 12, Line 259, "a significant increasing trend of: : :": is this "a significant increasing trend of total RA in temperate zones,: : :."??

Response: thank you for your careful revision. Yes, we mean "a significant increasing trend of total RA in temperate zones….". We revised the text:

"However, there were significant increasing trends of total RA in temperate zones, temperate forests, savannas and wetland of Hashimoto2015-RA, which were not observed in RF-RA".

Page13, Figure 9: I cannot understand the importance of this figure.

Response: We appreciate your question. Figure 9 showed the relative importance of three main

environmental drivers – MAT, MAP and SWR, by colors with RGB plot.

Due to different ecosystem types, or plant functional types or climate zones, the dominant factors may vary. As indicated by Fig. 10, 56% of land area was dominated by precipitation, while temperature and shortwave radiation dominated 19% and 25% of global land areas, which indicated an uneven control of environmental factors on RA. Therefore, Figure 9 showed the spatial variability of dominance of MAT, MAP and SWR on RA. It was found that the dominance of precipitation on RA was globally distributed, particularly dry or semi-arid areas, such as Northwest China, Southern Africa, Middle Australia and America, while temperature controlled RA mainly in in tropical Africa, Southern Amazon rainforests, Siberia and partly tundra, and shortwave radiation dominated high latitudinal areas, e.g. Eastern America and middle and Eastern Russian. Such analysis have been widely used in other studies, e.g. gross primary production (Yao et al., 2018), earth greening (Zhu et al., 2016), vegetation productivity (Seddon et al., 2016).

RGB synthesis (Fig. 9) was performed on stretched values of partial correlation coefficients, an effective way to illustrate the spatial distribution of dominant driving factors of RA (Yao et al., 2018), which could increase our understanding the mechanisms and spatial variability of environmental controls on RA at the global scale.

Page 14, Line 290 "For example, temperature was the: : :Australia" is that the result of Hashimoto et al.(2015)?

Response: thank you for your careful revision again. Yes, we mean "Hashimoto2015-RA", and revised in text:

"temperature was the main dominant factor for most areas in Australia for Hashimoto2015-RA".

References

Ballantyne, A., Smith, W., Anderegg, W., Kauppi, P., Sarmiento, J., Tans, P., Shevliakova, E., Pan, Y., Poulter, B., Anav, A., Friedlingstein, P., Houghton, R., and Running, S.: Accelerating net terrestrial carbon uptake during the warming hiatus due to reduced respiration, Nature Clim. Change, 7, 148-152, http://dx.doi.org/10.1038/nclimate3204, 2017.

Bond-Lamberty, B.: New Techniques and Data for Understanding the Global Soil Respiration Flux, Earth's Future, 6, 1176-1180, http://dx.doi.org/10.1029/2018ef000866, 2018.

Bond-Lamberty, B., Wang, C., and Gower, S. T.: A global relationship between the heterotrophic and autotrophic components of soil respiration?, Glob. Chang. Biol., 10, 1756-1766, http://dx.doi.org/10.1111/j.1365-2486.2004.00816.x, 2004.

Hanson, P. J., Edwards, N. T., Garten, C. T., and Andrews, J. A.: Separating root and soil microbial contributions to soil respiration: A review of methods and observations, Biogeochemistry, 48, 115-146, http://dx.doi.org/10.1023/a:1006244819642, 2000.

Harris, I., Jones, P., Osborn, T., and Lister, D.: Updated high-resolution grids of monthly climatic observations–the CRU TS3. 10 Dataset, Int. J. Climatol., 34, 623-642, http://dx.doi.org/10.1002/joc.3711, 2014.

Hashimoto, S., Carvalhais, N., Ito, A., Migliavacca, M., Nishina, K., and Reichstein, M.: Global spatiotemporal distribution of soil respiration modeled using a global database, Biogeosciences, 12, 4121–4132, http://dx.doi.org/10.5194/bgd-12-4331-2015, 2015.

Hengeveld, G. M., Gunia, K., Didion, M., Zudin, S., Clerkx, A. P. P. M., and Schelhaas, M. J.: Global 1-degree Maps of Forest Area, Carbon Stocks, and Biomass, 1950-2010. ORNL Distributed Active Archive Center, https://doi.org/10.3334/ORNLDAAC/1296, 2015.

Huang, J., Li, Y., Fu, C., Chen, F., Fu, Q., Dai, A., Shinoda, M., Ma, Z., Guo, W., Li, Z., Zhang, L., Liu, Y., Yu, H., He, Y., Xie, Y., Guan, X., Ji, M., Lin, L., Wang, S., Yan, H., and Wang, G.: Dryland climate change: Recent progress and challenges, Rev. Geophys., 55, 719-778, http://dx.doi.org/10.1002/2016rg000550, 2017.

Jung, M., Reichstein, M., Margolis, H. A., Cescatti, A., Richardson, A. D., Arain, M. A., Arneth, A., Bernhofer, C., Bonal, D., Chen, J. Q., Gianelle, D., Gobron, N., Kiely, G., Kutsch, W., Lasslop, G., Law, B. E., Lindroth, A., Merbold, L., Montagnani, L., Moors, E. J., Papale, D., Sottocornola, M., Vaccari, F., and Williams, C.: Global patterns of land-atmosphere fluxes of carbon dioxide, latent heat, and sensible heat derived from eddy covariance, satellite, and meteorological observations, J. Geophys. Res. Biogeosci., 116, G00J07, http://dx.doi.org/10.1029/2010jg001566, 2011.

Le Quéré, C., Andrew, R. M., Friedlingstein, P., Sitch, S., Pongratz, J., Manning, A. C., Korsbakken, J. I., Peters, G. P., Canadell, J. G., Jackson, R. B., Boden, T. A., Tans, P. P., Andrews, O. D., Arora, V. K., Bakker, D. C. E., Barbero, L., Becker, M., Betts, R. A., Bopp, L., Chevallier, F., Chini, L. P., Ciais, P., Cosca, C. E., Cross, J., Currie, K., Gasser, T., Harris, I., Hauck, J., Haverd, V., Houghton, R. A., Hunt, C. W., Hurtt, G., Ilyina, T., Jain, A. K., Kato, E., Kautz, M., Keeling, R. F., Klein Goldewijk, K., Körtzinger, A., Landschützer, P., Lefèvre, N., Lenton, A., Lienert, S., Lima, I., Lombardozzi, D., Metzl, N., Millero,

F., Monteiro, P. M. S., Munro, D. R., Nabel, J. E. M. S., Nakaoka, S.-i., Nojiri, Y., Padín, X. A., Peregon, A., Pfeil, B., Pierrot, D., Poulter, B., Rehder, G., Reimer, J., Rödenbeck, C., Schwinger, J., Séférian, R., Skjelvan, I., Stocker, B. D., Tian, H., Tilbrook, B., van der Laan-Luijkx, I. T., van der Werf, G. R., van Heuven, S., Viovy, N., Vuichard, N., Walker, A. P., Watson, A. J., Wiltshire, A. J., Zaehle, S., and Zhu, D.: Global Carbon Budget 2017, Earth Syst. Sci. Data, 10, 405-448, http://dx.doi.org/10.5194/essd-2017-123, 2018.

Raich, J. W., Potter, C. S., and Bhagawati, D.: Interannual variability in global soil respiration, 1980–94, Glob. Chang. Biol., 8, 800-812, doi:10.1046/j.1365-2486.2002.00511.x, 2002.

Seddon, A. W., Macias-Fauria, M., Long, P. R., Benz, D., and Willis, K. J.: Sensitivity of global terrestrial ecosystems to climate variability, Nature, 531, 229-232, http://dx.doi.org/10.1038/nature16986, 2016.

Yao, Y., Wang, X., Li, Y., Wang, T., Shen, M., Du, M., He, H., Li, Y., Luo, W., Ma, M., Ma, Y., Tang, Y., Wang, H., Zhang, X., Zhang, Y., Zhao, L., Zhou, G., and Piao, S.: Spatiotemporal pattern of gross primary productivity and its covariation with climate in China over the last thirty years, Glob. Chang. Biol., 24, 184-196, http://dx.doi.org/10.1111/gcb.13830, 2018.

Zhu, Z., Piao, S., Myneni, R. B., Huang, M., Zeng, Z., Canadell, J. G., Ciais, P., Sitch, S., Friedlingstein, P., Arneth, A., Cao, C., Cheng, L., Kato, E., Koven, C., Li, Y., Lian, X., Liu, Y., Liu, R., Mao, J., Pan, Y., Peng, S., Peñuelas, J., Poulter, B., Pugh, T. A. M., Stocker, B. D., Viovy, N., Wang, X., Wang, Y., Xiao, Z., Yang, H., Zaehle, S., and Zeng, N.: Greening of the Earth and its drivers, Nature Climate Change, 6, 791-795, http://dx.doi.org/10.1038/nclimate3004, 2016.

---

## Author Response (AR2)

**Response to editor**

Dear editor – Dr. Heim,

Great thanks for your really careful revision of our manuscript, and we strongly believe that your suggestion and comments will greatly increase the quality of our manuscript, which could be more fit to ESSD requirement.

According to your suggestion and comments, we have revised our manuscript carefully and thoroughly. Please see, below, our point-to-point response.

According to the journal requirement, we structured the response as: (1) comments from Referees, (2) authors' response (two referees and one editor), and (3) authors' changes in manuscript.

**Suggestion and comments from referees or editor** are marked in **Black**.

**Responses to referee and editor's comments** are labelled in **blue**.

**Cited changes made in the manuscript** are marked in **red**.

Please do not hesitate to let us know if you have additional questions and/or comments.

Sincerely,

Xiaolu Tang, Wenjie Zhang and Sicong Gao, on behalf of all co-authors.

Comments to the Author:

The published global time series on belowground autotrophic respiration is of importance. As the data description and data publication do not yet fulfill the requirements of ESSD and improvements are needed a major revision of the manuscript and a minor revision and new publication of the dataset is needed.

The manuscript provides a lot of detailed information and an interesting cross comparison with the Hashimoto et al. 2015 global data set on belowground respiration. However, the description of the data set, its data sources and the data generation is lost in the complexity of the paper. This happens because the article is not exclusively focused on the dataset but put also a focus on its application. Recommendations to the authors: for ESSD the focus of the manuscript should be on data sets and products. the word 'study' is frequently used and could sometimes be exchanged with 'data set'.

Response: great thanks to Dr. Birgit Heim, and your suggestion really improves the quality and qualification of our manuscript to ESSD. According to your suggestion and comments, we had a major revision our manuscript and focused more on the developed RH dataset, specifically:

**First**, in abstract, we added more technical details on developing the RA dataset:

"We applied Random Forest (RF) algorithm to upscale an updated dataset from Global Soil Respiration Database (v4) – covering all major ecosystem types and climate zones with 449 field observations, using globally gridded temperature, precipitation, soil and other environmental variables. We used a 10-fold cross-validation to evaluate the performance of RF to predict the spatial and temporal pattern of RA. Finally, a globally gridded RA dataset from 1980 to 2012 was produced with a spatial resolution of $0.5^{o} \times 0.5^{o}$ (longitude $\times$ latitude) and a temporal resolution of one year, expressed by g C m$^{-2}$ yr$^{-1}$ (gram carbon for per square meter per year)."

**Second**, "study" was changed to dataset in many cases as suggested:

For instance (just list some examples here):

"the outcome of **this study** will advance $\rightarrow$ The **developed RF-RA dataset** will advance …..."

"**Figure 1** Observational sites used **in this study** $\rightarrow$ **Figure 1** Distribution of observational sites used to develop the **globally gridded RF-RA dataset**"

"**this study** compared other RA …. $\rightarrow$ **the RF-RA dataset** was compared with other RA"

"RA estimates in **this study** fell in this range $\rightarrow$ The **developed RF-RA dataset** fell in this range"

**Third**, we replace "**this study**" in figures by "**RF-RA**", which was named by Random Forest method as suggested. E.g. Figure 3 and 7.

**Fourth**, in chapter 5, we included more details on data availability as:

"The developed RF-RA dataset is freely downloadable from https://doi.org/10.6084/m9.figshare.7636193 (Tang et al., 2019), named as "Respiration_autotrophic_belowgroud_glob_1980_2012_yr_half_dgree.nc", which is a globally gridded RA dataset from 1980 to 2012 with a spatial resolution of 0.5 degree at an annual scale, expressed by g C m$^{-2}$ yr$^{-1}$ (gram carbon for per square meter per year). The RA dataset is provided in Netcdf format (Network Common Data Form)."

**Fifth**, according to your suggestion, we also published a new version of dataset with more technical details and explanations. Please see: https://doi.org/10.6084/m9.figshare.7636193, and we revised as:

This data repository contains (1) yearly global autotrophic respiration (RA) dataset from 1980 to 2012 with a spatial resolution of 0.5°; (2) original field observations to develop Random Forest (RF) model; (3) main R codes to produce RA database.

Model description:

The globally gridded RA database was developed by Random Forest (RF) with 449 field observations (see "dataset.csv" in this repository, updated from Bond-Lamberty and Thomson, 2018) using 11 global variables, including gridded temperature, precipitation, diurnal temperature range, potential evapotranspiration, Palmer Drought Severity Index, nitrogen deposition, downward shortwave radiation, soil carbon content, soil nitrogen density, soil water content, land cover.

Dataset information:

Dataset name: "Respiration_autotrophic_belowgroud_glob_1980_2012_yr_half_dgree.nc"

Which means globally belowground autotrophic respiration from 1980 to 2012 with a spatial resolution of 0.5° at a yearly step.

Units: g C m$^{-2}$ yr$^{-1}$

Format: network Common Data Form (netCDF)

Spatial coverage: 90S-90N, 180W-180E

The "dataset.csv" file is the field observation from peer review publications combining Global Soil Respiration Database (SRDB v4, Bond-Lamberty and Thomson, 2018), which is publicly available at https://daac.ornl.gov/cgi-bin/dsviewer.pl?ds_id=1578. Besides, The database was further updated using observations collected from the China Knowledge Resource Integrated Database (www.cnki.net) up to November 2018 according to the criteria of SRDB. This dataset is provided in format of ".csv".

R codes:

10fold_CV_RA.txt: 10-fold CV for RA

Annual_variability_RA.txt: annual variability for global RA

CMP_RA.txt: comparing RF-RA and Hashimoto2015-RA using CMP approach

Ra_DD_CC_plot.txt: plotting the comparing results from CMP

RA_MAT_MAP_anomaly.txt: plotting and modelling the relationship between temperature/precipitation anomalies and RA

RGB_plot.txt: deriving RGB plot to detecting the relative importance of temperature, precipitation and shortwave radiation.

Requirements for publication:

1) data sources need to be cited correctly

I) It is not clear which version of the Global data base was used: in the Introduction it is cited as "from the most updated global soil respiration dataset (SRDB v4, doi: 10.5194/bg-7-1915-2010, Bond-Lamberty and Thomson, 2010a) "– a formally better citation would be: Global Database of Soil Respiration Data Version 4 (SRDB v4, Bond-Lamberty and Thomson, 2010a) However the manuscript states on p.18, L407… this study used an up-to-date field observational database developed from SRDB up to the end 2018. Does this refer to this current version (2018)? Bond-Lamberty, and Thomson. 2018. A Global Database of Soil Respiration Data, Version 4.0. ORNL DAAC, Oak Ridge, Tennessee, USA. https://doi.org/10.3334/ORNLDAAC/1578

Response: sorry for the inconsistent citations. We used the most updated SRDB dataset up to November 2018, so we cited "Bond-Lamberty, and Thomson. 2018. A Global Database of Soil Respiration Data, Version 4.0. ORNL DAAC, Oak Ridge, Tennessee, USA. https://doi.org/10.3334/ORNLDAAC/1578" though the manuscript.

In chapter 2.1, p.5 L121, details on "Q10", "Q11", "Q12", "Q13" and "Q14" should be given

Response: more details were given according to the original SRDB (Bond-Lamberty and Thomson, 2018) as follows:

"Q10" (potential problem with data), "Q11" (suspected problem with data), "Q12" (known problem with data), "Q13" (duplicate?) and "Q14" (inconsistency)

II) the gridded input data sets are not correctly referenced

Examples: p.6, L136 "soil nitrogen content was downloaded from Spatial Data Access Tool (https://webmap.ornl.gov/ogc/index.jsp)" is referring to a web visualization and download service of this data set. However, the information needed for this input variable is it's correct naming and the correct citation of Authors, or a Program. In this case the variable name is 'Granule soil total nitrogen density', the data set collection: 'Global Soil Data Task', the Program 'International Geosphere-

Biosphere Programme (IGBP)', the year of the first publication of this dataset: 2000.

Response: as suggested, we first move the table S1 to the main text as "Table 1 Global variables used for producing the global RH dataset".

In the modified Table 1, we first provided the download link and the cite the references as "author + year", or "program + year" as follows:

**Table 1** Global variables used for producing the global RH dataset

| | | Variables | Type | Type of variability | Sources |
|---|---|---|---|---|---|
| Climate | | Mean annual temperature (°C) | Split | Yearly | https://crudata.uea.ac.uk/cru/data/hrg/cru_ts_4.01/ (Harris et al., 2014) https://crudata.uea.ac.uk/cru/data/drought/ (van der Schrier et al., 2013) |
| | | Mean annual precipitation (mm) | Split | Yearly | |
| | | Diurnal temperature range (°C) | Split | Yearly | |
| | | Potential evapotranspiration (mm) | Split | Yearly | |
| | | Palmer Drought Severity Index | Split | Yearly | |
| | | Nitrogen deposition (g N m$^{-2}$ yr$^{-1}$) | Split | Yearly | https://www.isimip.org/gettingstarted/availability-input-data-isimip2b/ (Lamarque et al., 2013) |
| | | Downward Shortwave radiation (W m$^{-2}$) | Split | Yearly | ftp://ftp.cdc.noaa.gov/Datasets/ncep.reanalysis.derived/surface_gauss/dswrf.sfc.mon.mean.nc (Kalnay et al., 1996) |
| Soil | | Soil carbon content (g kg$^{-1}$) | - | Static | https://soilgrids.org/#!/?layer=TAXNWRB_250m (Hengl et al., 2017) |
| | | Soil nitrogen density (g m$^{-2}$) | - | Static | https://webmap.ornl.gov/ogc/dataset.jsp?ds_id=569 (Global Soil Data, 2000) |
| | | Soil water content (mm) | Split | Yearly | https://www.esrl.noaa.gov/psd/data/gridded/data.cpcsoil.html (van den Dool et al., 2003) |
| Land cover | | MODIS land cover | - | Static | https://glcf.umd.edu/data/lc/ (Friedl et al., 2010) |

A DOI was assigned in 2014 that should be shown in the reference list.

Response: DOI was expressed in reference list, and we cited the most updated reference as suggested above.

The authors should provide in 2.2. a table with more details as it is now displayed in the supplement, with correct references and units (e.g., weight% of total nitrogen).

Response: we moved Table S1 from supplement to main text, see "**Table 1** Global variables used for producing the global RH dataset" as Table 1 above. The units were given as well.

Most probably all the gridded time series data used as input in this study are constrained onto the time period 1980 to 2012?

Response: yes, all variables listed in Table 1 labelled by "Yearly" were constrained on the time period from 1980 to 2012.

all abbreviations needed to be spelled out when they are used the first time in the text: e.g. CRU TS, NOAA/ESRL etc.

Response: done! CRU TS means Climatic Research Unit time-series, while NOAA/ESRL stands for National Oceanic and Atmospheric Administration/Earth System Research Laboratory. Revised in the text as follows:

"Monthly gridded temperature, precipitation, diurnal temperature range, potential evapotranspiration, and self-calibrated Palmer Drought Severity Index (PDSI) at 0.5º resolution were obtained from Climatic Research Unit (CRU) time-series (TS) Version 4.01 from 1901 to 2016 (Harris et al., 2014; van der Schrier et al., 2013). Monthly shortwave radiation (SWR, Kalnay et al., 1996),  and soil water content (van den Dool et al., 2003) at 0.5º resolution were from National Oceanic and Atmospheric Administration/Earth System Research Laboratory (NOAA/ESRL) at Physical Sciences Division"

2) Suggestion 2.6 to provide a more specific subtitle e.g. 'cross comparison with Hashimoto 2015 Soil Respiration' The authors need to provide details on the Hashimoto et al 2015 product, a short overview on key points how it was produced (to avoid that the readers of this manuscript need to read Hashimoto et al. 2015 to understand the data set production and pelicularities) and how the data set was treated for further analyses.

Example on the details of input data and the methods that Hashimoto et al 2015 used for the generation of their product are required, e.g, that gridded air temperature and precipitation data were taken from East Anglia Climatic Research Unit CRU, soil respiration data from SRDB, Bond-Lamberty and Thomson, 2010

How is the product autotrophic respiration from the Hashimoto et al 2015 product derived (Hashimoto et al product representing the total soil respiration) – this retrieval step is so far not described as a processing that it is undertaken by the author team.

Response: first the subtitle was changed to "Cross comparison with Hashimoto2015-RA". The details on how to develop Hashimoto RA, including variables, models, and calculating RA, were described in main text:

"Hashimoto developed a climate-driven model by updating Raich's model, which stimulated soil respiration as a function of temperature and water (precipitation) at a monthly time step (Hashimoto et al., 2015; Raich et al., 2002). Therefore, to get a global estimate to soil respiration at a monthly scale, the globally gridded air temperature and precipitation with a spatial resolution of 0.5º were derived from University of East Anglia CRU 3.21 (Harris et al., 2014), and 1638 field observations were taken from SRDB (v3) for model parameterization (Hashimoto et al., 2015). Monthly soil

respiration was summed to a yearly scale. Furthermore, annual soil respiration was divided into autotrophic and heterotrophic respiration using a global relationship between soil respiration and heterotrophic respiration derived from a meta-analysis (Bond-Lamberty et al., 2004). This global relationship can be expressed by:

$$\ln(RH) = 1.22 + 1.73 \times \ln(RS) \qquad (1)$$

Where RH means annual heterotrophic respiration, and RS stands for annual soil respiration, expressed by g C m$^{-2}$ yr$^{-1}$.

Therefore, global Hashimoto2015-RA was derived by the difference between soil respiration and heterotrophic respiration. The monthly or annual Hashimoto2015-RA dataset can be freely accessed from (http://cse.ffpri.affrc.go.jp/shojih/data/index.html, Hashimoto et al., 2015)."

the product from Hashimoto et al 2015 should always be named together with the publication year, or shorter for Figures etc. as Hashimoto 2015. The naming of the Random Forest driven data set as 'data driven product' is still a very general description, one could construct a name using the Random Forest method, and also take this name instead of referring in figures to 'this study'.

Response: as suggested, "Hashimoto RA" was changed to "Hashimoto2015-RA", while the developed RA dataset was named as "RF-RA" throughout the manuscript and figures, e.g. Figure 3 and 7.

3) For publication in ESSD the authors need to optimize the metadata of their product and publish the new version on figshare.

I) -global attributes: should contain a meta data field with the name of the product;

-variable: the long_name should contain the name of the variable, e.g. autotrophic respiration, the unit should contain 'C m-2 y-1';

-z: the long_name should contain more details on z, the unit should contain 'years since 1980'

file name is now Ra.10Fold.CV.720.360.1980.2012. Consider to change the file name to a less technically-focused file name, consider more explaining keywords, e.g. Respiration_autotroph_glob-author_year.

Response: we publish a new version of developed RA dataset with more details as follows (as well at https://doi.org/10.6084/m9.figshare.7636193):

This data repository contains (1) yearly global autotrophic respiration (RA) dataset from 1980 to 2012 with a spatial resolution of 0.5°; (2) original field observations to develop Random Forest (RF) model; (3) main R codes to produce RA database.

Model description:

The globally gridded RA database was developed by Random Forest (RF) with 449 field observations (see "dataset.csv" in this repository, updated from Bond-Lamberty and Thomson, 2018) using 11 global variables, including gridded temperature, precipitation, diurnal temperature range, potential evapotranspiration, Palmer Drought Severity Index, nitrogen deposition, downward shortwave radiation, soil carbon content, soil nitrogen density, soil water content, land cover.

Dataset information:

Dataset name: "Respiration_autotrophic_belowgroud_glob_1980_2012_yr_half_dgree.nc"

Which means globally belowground autotrophic respiration from 1980 to 2012 with a spatial resolution of 0.5° (WGS84 ellipsoid) at an annual step.

Units: g C m$^{-2}$ yr$^{-1}$

Format: network Common Data Form (netCDF)

Spatial coverage: 90S-90N, 180W-180E

The "dataset.csv" file is the field observation from peer review publications combining Global Soil Respiration Database (SRDB v4, Bond-Lamberty and Thomson, 2018), which is publicly available at https://daac.ornl.gov/cgi-bin/dsviewer.pl?ds_id=1578. Besides, The database was further updated using observations collected from the China Knowledge Resource Integrated Database (www.cnki.net) up to November 2018 according to the criteria of SRDB. This dataset is provided in format of ".csv".

R codes:

10fold_CV_RA.txt: 10-fold CV for RA

Annual_variability_RA.txt: annual variability for global RA

CMP_RA.txt: comparing RF-RA and Hashimoto2015-RA using CMP approach

Ra_DD_CC_plot.txt: plotting the comparing results from CMP

RA_MAT_MAP_anomaly.txt: plotting and modelling the relationship between temperature/precipitation anomalies and RA

RGB_plot.txt: deriving RGB plot to detecting the relative importance of temperature, precipitation and shortwave radiation.

Additionally, the dataset title was changed to "**A gridded dataset of belowground autotrophic respiration from 1980 to 2012 in global terrestrial ecosystems upscaling of field observations**" to attract the readers and users.

II) Abstract and title on figshare

Now it reads 'Global variability of belowground autotrophic respiration in terrestrial ecosystems' ..the netcdf file represent the global belowground autotrophic respiration from 1980 to 2012 with a resolution of half degree. The unit is g C m-2 a-1 the product was produced by global published observations with the linkage of global climate soil and other environmental variables using random forest.'

Suggestion to change the title more towards a product (it does not need to be the same title as in ESSD)

e.g. title: Global belowground autotrophic respiration from 1980 to 2012

Response: we changed the title of dataset in Figshare to: "A gridded dataset of belowground autotrophic respiration from 1980 to 2012 in global terrestrial ecosystems upscaling of field observations", see https://doi.org/10.6084/m9.figshare.7636193.

abstract text – suggestion to put more technical details in, some suggestions what information in the figshare abstract can be provided below

global belowground autotrophic respiration from 1980 to 2012 …. Global Database of Soil Respiration Data (Bond-Lamberty and Thomson…), …environmental climate and soil variables … random forest. …The unit is C m-2 y-1'. The spatial resolution is 0.5 degree latitude longitude on WGS84 ellipsoid (?), temporal resolution yearly …

Response: in abstract, we added more technical details as:

"We applied Random Forest (RF) algorithm to upscale an updated database of 449 field observations from Global Soil Respiration Database (v4) – covering all major ecosystem types and climate zones, using global gridded temperature, precipitation, soil and other environmental variables. The performance of RF was evaluated by a 10-fold cross-validation. Finally, a globally gridded RA dataset from 1980 to 2012 was produced with a spatial resolution of 0.5 degree and a temporal resolution of one year, expressed by g C m$^{-2}$ yr$^{-1}$ (gram carbon for per square meter per year)."

In figshare, we added further details as suggested. Please see detailed response of comments **"3)-II"**

4) chapter 5 data availability

Describe here details of the data set taken from the figshare abstract, e.g. that it is a global data set with x bands the variable yearly autotrophic respiration unit C m-2 y-1'

Response: done as suggested:

"The developed dataset is freely downloadable from https://doi.org/10.6084/m9.figshare.7636193 (Tang et al., 2019), named as "Respiration_autotrophic_belowgroud_glob_1980_2012_yr_half_dgree.nc", which means a

globally gridded RA dataset from 1980 to 2012 with a spatial resolution of 0.5 degree at an annual scale, expressed by g C m$^{-2}$ yr$^{-1}$ (gram carbon for per square meter per year)". The RA dataset is provided in Netcdf format (Network Common Data Form).

Non-public comments to the Author:

you got a major revision.

The major editorial request is to provide throughout information on all the data sources and to edit the data product and its publication on figshare

One interesting component of your manuscript is the comparison of spatiotemporal patterns with Hashimoto 2015

Please enlarge Figures 2, 4, 5

Response: done, and see changes in text.

Enlarge figures 9 so that both a and b figures are displayed across the full paragraph width

Response: done, and see changes in text.

Since the figure quality was reduced when convert word to pdf, we will attach the original figures, which are in pdf format, once the manuscript is accepted.

Discussion

Despite the major drivers are MAT, MAP, SWR that are all atmospheric variables your input into the random forest are also the gridded data on land cover and soil units and this is visible in the spatial patterns of your output product autotrophic respiration, e.g. there are some relative sharp boundaries visible, major land cover units are outlined. Do you recognize some major spatial patterns in the spatial pattern of the gridded input land cover and soil variables? do land cover and soil gridded data display similar spatial patterns?

Response: to represent ecosystem type as an important variable driving RA, we include ecosystem type to develop RF model. Figure 1 shows the dominant driving factors for RF-RA and the distribution of MODIS global land cover. Comparing the figure 1a, 1b, 1c and 1d, we did not see sharp boundaries for major land covers. Soil gridded data and land cover data did not show similar patterns.

[Figure]

Figure 1 The dominant driving factors for RF-RA (a), the global distribution of land cover types (b) and soil carbon content (c).

Supplement

Table 1: Please move the overview on input data into the main manuscript providing correct citation of the sources, see also public comment

Response: we have moved to Table S1 to main text (Table 1) and provided more details with download link and citations by (author + years) as suggested.

Fig S7 would also fit in the main manuscript

Response: Fig. S7 was moved to the main manuscript as:

"**Figure 10** The percentage of land areas (calculated from cell areas) dominated by mean annual temperature (MAT), precipitation (MAP) and shortwave radiation (SWR) for RF-RA and Hashimoto2015-RA."

Since the figure quality was reduced when convert word to pdf, we will attach the original figures, which are in pdf format, once the manuscript is accepted.

Fig S8 not needed, citation of values in the main manuscript

Response: this figure was removed.

**Response to Reviewer #1**

Dear editor and reviewer,

Thank you very much for your great efforts, comments and suggestion again! Based on the first version of "the response to reviewer", the editor evaluated our manuscript carefully and proposed some good suggestion and comments. Combining editor and reviewers' suggestion and comment, we revised our manuscript carefully and thoroughly.

Therefore, I am kindly to remind you that it would be much more efficient to work on **the updated** "the response to reviewer" and **discard the old version that uploaded on July 17, 2019.** Thank you for your understanding.

**Suggestion and comments from referees or editor** are marked in **Black**.

**Responses to referee and editor's comments** are labelled in **blue**.

**Cited changes made in the manuscript** are marked in **red**.

Please do not hesitate to let us know if you have further questions and/or comments.

Sincerely,

Xiaolu Tang, Wenjie Zhang and Sicong Gao, on behalf of all co-authors.

Response to reviewer #1

This manuscript deals with estimation of global belowground autotrophic respiration (RA) in terrestrial ecosystems. I have some questions in this study. 1) Authors compared global RA by data-derived with that by Hashimto et al.(2015). However, authors did not refer the data of Hashimoto et al. (2015) in the manuscript. How did authors get the data from Hashimoto? Please explain the difference between Random forest model and methods of Hashimoto et al. (2015).

Response: we apologize for the unclear statement of Hashimoto RH. To avoid that the readers of this manuscript need to read Hashimoto et al. 2015 to understand the RA dataset, we first added a short summary about the details of input data, the method used and how soil respiration and RA derived. Then we provided details on how to get Hashimoto2015-RA (renamed as suggested by editor to fit ESSD better).

Detailed in text as:

"Hashimoto developed a climate-driven model by updating Raich's model, which stimulated soil respiration as a function of temperature and water (precipitation) at a monthly time step (Hashimoto et al., 2015; Raich et al., 2002). Therefore, to get a global estimate to soil respiration at a monthly scale, the globally gridded air temperature and precipitation with a spatial resolution of 0.5º were derived from University of East Anglia CRU 3.21 (Harris et al., 2014), and 1638 field observations were taken from SRDB (v3) for model parameterization (Hashimoto et al., 2015). Monthly soil respiration was summed to a yearly scale. Furthermore, annual soil respiration was divided into autotrophic and heterotrophic respiration using a global relationship between soil respiration and heterotrophic respiration derived from a meta-analysis (Bond-Lamberty et al., 2004). This global relationship can be expressed by:

$$\ln(RH) = 1.22 + 1.73 \times \ln(RS) \qquad (1)$$

Where RH means annual heterotrophic respiration, and RS stands for annual soil respiration, expressed by g C m$^{-2}$ yr$^{-1}$.

Therefore, global Hashimoto2015-RA was derived by the difference between soil respiration and heterotrophic respiration. The monthly or annual Hashimoto2015-RA dataset can be freely accessed from (http://cse.ffpri.affrc.go.jp/shojih/data/index.html, Hashimoto et al., 2015)."

Therefore, the main differences between Random Forest based RA (RF-RA) and Hashimoto20105-RA included: (1) variables used to develop models. To develop RF-RA, we applied 11 variables, including temperature, precipitation, soil nitrogen, soil carbon and other environmental variables, while on

temperature and precipitation were applied in develop Hashimoto2015-RA; (2) modelling approaches. We used Random Forest algorithm to model RA, while a simple climate-driven model by temperature and precipitation was applied to develop Hashimoto2015-RA. Consequently, we obtained a much higher model efficiency (52%) compared to Hashimoto2015-RA (32%). Furthermore, Random Forest algorithm have great potentials to address the non-linear correlation between RA and environmental variables, and remove auto-correlations among environmental variables.

2) Authors used PgC a-1 or gC m-2 a-1for the unit of RA, but I guess that a-1 should be yr-1. Please correct all unit in the manuscript and figures.

Response: yes, it means Pg C per year. Corrected to "Pg C yr$^{-1}$" or "g C m$^{-2}$ yr$^{-1}$" throughout the manuscript and figures!

3) Authors discussed about importance of the dominant environmental factors for estimate spatio-temporal variation in RA. I think that it is important not only environmental factors for plant production but also plant biomass because root respiration would have positive correlation with plant biomass. Why did authors ignore the global pattern of plant biomass??

Response: thank you for the good comments. We agree with you that plant biomass, particularly root biomass, would have positive correlations with RA. However, selecting variables is constrained by the fact that a variable must be available at all sites and at the corresponding global product simultaneously. For instance, if a variable is measured accurately at sites, but with large uncertainties in the corresponding global product, it may be advantageous to exclude this variable from the analysis (Jung et al., 2011).

Although we tried to include global plant or root biomass as a driving variable, we found such product was only available for a single year, or mean values of several years (Huang et al., 2017), or forests (Hengeveld et al., 2015), and there was a lack of time-series global biomass product covering all land covers. Given the fact that plant biomass was highly dynamic due to annual accumulation, using a global biomass for a given year or particularly ecosystem type to represent the biomass dynamics covering all terrestrial ecosystems would cause a great uncertainty to RA estimation. Therefore, the lack of global biomass product constrained the use of plant biomass as a driving variable for RA in this study. Instead, we used MODIS land cover as one of driving variables, which could indirectly reflect the biotic or biomass control on RA to some extent.

Finally, please considering my specific comments and get some English proofreading. In addition, please

reconsider carefully about all figures, because I feel that some figures are not important in this manuscript. If authors resolve these questions, I think that this manuscript would be better for global data science.

Response: we answered each of your specific comment carefully, and we improved the English carefully.

As you suggested, see specific comments below, Figure 6c was not important and removed.

Specific comments

Page 3, line 54, "which is almost 5 times of: : :.": I cannot understand relationship between this sentence and preceding sentence. Page 3, line 56, "Therefore, an accurate estimate of : : :": I think that authors did not enough explain the reasons before the sentences. Please add more explanation.

Response: Since the two comments link with each other, we answer the two comments together.

We apologize for the unclear statement. We revised and added more explanation for it as follows:

"Globally, RA could amount roughly up to 54 Pg C yr$^{-1}$ (1 Pg = $10^{15}$ g, calculating RA as an approximate ratio of 0.5 of soil respiration, more details in Hanson et al., 2000) according to different estimates of global soil respiration (Bond-Lamberty, 2018), which is almost 5 times of the carbon release from human activities (Le Quéré et al., 2018). However, the contribution of RA to soil respiration varied greatly from 10% to 90% across biomes, climate zones and among years (Hanson et al., 2000), leading to the strong spatial and temporal variability in RA. Thus, whether RA varies with ecosystem types or climate zones remains an open question at the global scale (Ballantyne et al., 2017). Consequently, an accurate estimate of RA and its spatial-temporal dynamics are critical to understand the response of terrestrial ecosystems to climate change."

Page 8, Figure 2: I cannot understand the meaning of the figure 2c and 2b. Why did authors indicate the standard deviation of temporal variation in RA??

Response: Fig. 2b is the mean value of Hashimoto RA over 1980-2012, while Fig. 2c represents the standard deviation of predicted RA in this study. The figure caption was revised combining the other and editor suggestion and comments:

"**Figure 2** Spatial patterns of annual mean and standard deviation for RF-RA (a, c) and Hashimoto2015-RA (b, d) from 1980 to 2012, respectively. The standard deviation was applied to characterize the inter-annual variability following Yao et al. (2018)."

Due to the inter-annual variability of environmental controls on RA, RA varied annually. Although Fig. 6 describes the annual variability of total RA, the spatial pattern of annual variability of RA is lacking. To characterize the spatial pattern of annual variability of RA, the standard deviation of RA from 1980-2012 was employed. Such analysis was also conducted in other studies, e.g. Yao et al. (2018). Therefore, we used standard deviation to represent the temporal pattern of RA.

Page 11, Figure6: what the difference of Fig.6a and Fig.6b? Please add more explanation. And, please make the same value of yaxis in both of Fig.6a and Fig.6b. And I think that Fig6c is not needed.

Response: Fig. 6a represents the annual variability of predicted RA in this study, while Fig. 6b represents the annual variability of Hashimoto RA. The same value of yaxis from $39 - 45$ Pg C $yr^{-1}$ was applied.

Fig. 6c was not important and removed.

We corrected the figure description more clearly:

"Annual variability of RF-RA (a) and Hashimoto2015-RA (b) from 1980 to 2012. The grey area represents 95% confidence interval."

Page 12, Line 254 to 227, "All the biomes, except: : :, respectively": please rewrite these sentences. Grammatical subject is RA, I think.

Response: we rewrite these sentences:

"RA showed a significantly increasing trend during 1980-2012 ($p_s < 0.01$) in majority biomes, except temperate forests, savannas and wetland. RA in tropical forests, boreal forests and cropland increased by $0.0076\pm0.0015$, $0.0047\pm0.0016$, $0.0036\pm0.0014$ Pg C $yr^{-2}$, respectively."

Page 12, Line 259, "a significant increasing trend of: : :": is this "a significant increasing trend of total RA in temperate zones,: : :."??

Response: thank you for your careful revision. Yes, we mean "a significant increasing trend of total RA in temperate zones….". We revised the text:

"However, there were significant increasing trends of total RA in temperate zones, temperate forests, savannas and wetland of Hashimoto2015-RA, which were not observed in RF-RA".

Page13, Figure 9: I cannot understand the importance of this figure.

Response: We appreciate your question. Figure 9 showed the relative importance of three main

environmental drivers – MAT, MAP and SWR, by colors with RGB plot.

Due to different ecosystem types, or plant functional types or climate zones, the dominant factors may vary. As indicated by Fig. 10, 56% of land area was dominated by precipitation, while temperature and shortwave radiation dominated 19% and 25% of global land areas, which indicated an uneven control of environmental factors on RA. Therefore, Figure 9 showed the spatial variability of dominance of MAT, MAP and SWR on RA. It was found that the dominance of precipitation on RA was globally distributed, particularly dry or semi-arid areas, such as Northwest China, Southern Africa, Middle Australia and America, while temperature controlled RA mainly in in tropical Africa, Southern Amazon rainforests, Siberia and partly tundra, and shortwave radiation dominated high latitudinal areas, e.g. Eastern America and middle and Eastern Russian. Such analysis have been widely used in other studies, e.g. gross primary production (Yao et al., 2018), earth greening (Zhu et al., 2016), vegetation productivity (Seddon et al., 2016).

RGB synthesis (Fig. 9) was performed on stretched values of partial correlation coefficients, an effective way to illustrate the spatial distribution of dominant driving factors of RA (Yao et al., 2018), which could increase our understanding the mechanisms and spatial variability of environmental controls on RA at the global scale.

Page 14, Line 290 "For example, temperature was the: : :Australia" is that the result of Hashimoto et al.(2015)?

Response: thank you for your careful revision again. Yes, we mean "Hashimoto2015-RA", and revised in text:

"temperature was the main dominant factor for most areas in Australia for Hashimoto2015-RA".

**Response to Reviewer #2**

Dear editor and reviewer,

Thank you very much for your great efforts, comments and suggestion again! Based on the first version of "the response to reviewer", the editor evaluated our manuscript carefully and proposed some very good suggestion and comments. Combining editor and reviewers' suggestion and comments, we revised our manuscript carefully and thoroughly.

Therefore, I am kindly to remind you that it would be much more efficient to work on **the updated** "the response to reviewer" and **discard the old version that uploaded on July 17, 2019.** Thank you for your understanding.

**Suggestion and comments from referees or editor** are marked in **Black**.

**Responses to referee and editor's comments** are labelled in **blue**.

**Cited changes made in the manuscript** are marked in **red**.

Please do not hesitate to let us know if you have further questions and/or comments.

Sincerely,

Xiaolu Tang, Wenjie Zhang and Sicong Gao, on behalf of all co-authors.

Response to Reviewer #2

I have read "Global variability of belowground autotrophic respiration in terrestrial ecosystems". In the manuscript, the authors estimated global belowground autotrophic respiration from 1980-2012, analyzed the temporal trend, and explored the dominant factors for autotrophic variability. Global autotrophic respiration is a big carbon exchange between the atmosphere and terrestrial, but was rarely studies in the past years. Global temporal and spatial variability of autotrophic respiration is clearly a timely and interesting topic. Generally, this manuscript is well organized and easy to follow. The results and conclusions are reasonable. The production (Global belowground autotrophic respiration shared in the figShare) is a contribution to the community and potentially can serve as a benchmark for ecosystem models, it will be useful also make the analysis (include the codes) public available to make the analysis reproducible. But I think the authors have to better address the limitation, weakness, and uncertainty of this study. In my opinion, some major limitation including: 1) The sample size of RA: there are much less annual RA comparing with annual Rs (less than 10%), even though the authors extended the RA dataset by new papers from China Knowledge Resource Integrated (CNKI) Database, the total samples is only 449. And the majority of the samples are from the forest, samples from wetland and shrubland are extremely lacking (only 5 observations).

Response: we also attached the dataset and the R codes to generate the main results to figshare at https://doi.org/10.6084/m9.figshare.7636193.

Based on SRDB v4, including new observations from CNKI, we got a total of 4276 observations for soil respiration, however, there were 697 observations for RA. According to our selecting criteria: e.g. RA measurement lasting for one year; excluding measurements with Alkali absorption and soda lime; no site management, we got a RA dataset of 449 observations. Our observational dataset is mainly from forests, but a lack of observations in wetland and shrubland, which could be the limitation in this study. However, our dataset covered all major ecosystem types and climate zones across the globe.

We have discussed the limitation in "**4.4 Advantages, limitations and uncertainties**" section as follows:

"Finally, uneven coverage of observations in the updated dataset would be another source of uncertainties. Although our dataset had a wide range of land cover, the observational sites mainly distributed in China, Europe and North America and were dominated by forests. There was a lack of observations in areas, such as Africa, Austria and Russia, and biomes, such as tropical forest, shrubland, wetland and cropland. However, our dataset covered all major ecosystem types and climate zones across the globe. RA

observations caused bias of RF model toward the regions with more observations. Therefore, including more observations in these areas and biomes without observations should largely increase our capability to assess the spatial and temporal patterns of global RA and contribute to improve the global carbon cycling modelling to future climate change."

2) How can you evaluate the quality of the RA data? Even though the authors conducted quality control on the RA data, but it does not guarantee the reliability of the RA data. We lack reliable methods to separate RA and RH, current ways (e.g., trend, gap, girdling, clip, and isotope) have their own problem. Further, usually RH is measured, and RA was calculated as the difference between RS and RH, which also bring uncertainties. All those issues were not addressed and discussed in the manuscript. If the data reliability cannot be guaranteed, the estimates, trend, and dominant factors should also be questioned. Despite the above problems, I still think this study tend to address an important topic and may inspire more research in the future.

Response: we evaluate the quality of RA from different aspects to guarantee the reliability of RA: (1) measuring approaches: Alkali absorption and soda lime were not included due to the potential underestimate of respiration rate with the increasing pressure inside chamber (Pumpanen et al., 2004); (2) data quality control by quality flag: Q01 (estimated from figure), Q02 (data from another study), Q03 (data estimated-other), Q04 (potentially useful future data), Q10 (potential problem with data), Q11 (suspected problem with data), Q12 (known problem with data), Q13 (duplicate?), Q14 (inconsistency). Therefore, RA or total soil respiration observations labelled by "Q10", "Q11", "Q12", "Q13" and "Q14" were removed in this study. More details on data quality controls can be found in Bond-Lamberty and Thomson (2010a).

We agree with you that there was a lack reliable method to separate RA and RH, and current ways (e.g., trend, gap, girdling, clip, and isotope) have their own problems.

We have discussed the data quality and limitation of unreliable method to separate RA and RH in "**4.4 Advantages, limitations and uncertainties**" as follows:

"First, although we conducted a data quality control to develop the RF-RA dataset, a lack of reliable approach to separate RA and heterotrophic respiration may lead to an important uncertainty of RA estimates. There are several approaches, e.g. trenching, stable or radioactive isotope, gridding, to partition soil respiration (Bond-Lamberty et al., 2004; Högberg et al., 2001; Hanson et al., 2000), however, each of these approaches has its own limitations. For example, trenching has been widely applied to partition

RA and heterotrophic respiration due to easy operation and low cost, on the other hand, heterotrophic respiration may be increased due to the termination of water uptake by roots and the decomposition of remaining dead roots in trenching plots (Hanson et al., 2000; Tang et al., 2016). Commonly, RA was calculated from the difference between total soil respiration and heterotrophic respiration, thus the trenching approach might lead to an underestimation of RA. In our dataset, a total of 254 RA observations were estimated by trenching approach, while the rest RA observations were estimated by other separation approaches, e.g. isotope, radiocarbon, mass balance. Thus, inconsistent separation approaches could also be another source of uncertainty of RA values. ''

Specific comments Abstract

Line 22: (srdb v4) but later (line 97) you used (srdb version 4), be consistent.

Response: done!

Line 24: the unit for RA increasing trend should be Pg C a-2? Please see this paper: Ballantyne, A., Smith, W., Anderegg, W., Kauppi, P., Sarmiento, J., Tans, P., Shevliakova, E., et al. (2017). the warming hiatus due to reduced respiration. Nature Climate Change, 7(2), 148. https://doi.org/10.1038/NCLIMATE3204 – 152.

Response: thank you for your kind recommendation, and we corrected the increasing unit to Pg a $yr^{-2}$ or g C $m^{-2}$ $yr^{-2}$ throughout the text and figures, e.g. Figure 5.

Line 31-32: "the perspective that the parameters of global carbon stimulation independent on climate zones and biomes". But already some studies said that the response of respiration to climate change differs in different regions. Huang, Jian-ping, Xiao-dan Guan, and Fei Ji. "Enhanced cold-season warming in semi-arid regions." Atmospheric Chemistry and Physics 12.12 (2012): 5391-5398. Jian, Jinshi, et al. "Future global soil respiration rates will swell despite regional decreases in temperature sensitivity caused by rising temperature." Earth's Future 6.11 (2018): 1539-1554. The response of respiration to climate differs in different periods: Ballantyne, A., Smith, W., Anderegg, W., Kauppi, P., Sarmiento, J., Tans, P., Shevliakova, E., et al. (2017). the warming hiatus due to reduced respiration. Nature Climate Change, 7(2), 148. https://doi.org/10.1038/NCLIMATE3204 –152.

Response: thank you for your kind recommendation. Huang et al. (2012) mainly discussed the uneven changes of temperature, not RA.

Jian et al. (2018) found uneven changes of soil respiration in different areas, and Ballantyne et al. (2017)

also proposed that belowground autotrophic respiration may be varied among ecosystem types. These references have been cited to support our conclusions, and revised in the text as follows:

"However, RA increment varied with climate zones and ecosystem types (Figs. S2 and S3), which was similar to previous findings that total soil respiration or RA varied with climate zones or ecosystem types (Ballantyne et al., 2017; Jian et al., 2018a). These differences may be related to regional heterogeneity and plant functional trait. For example, regional temperature significantly differed from global averages (Huang et al., 2012), with much faster change in high-latitude regions (Hartmann et al., 2014), and semi-arid dominated the trend and variability of global land $CO_2$ sink (Ahlström et al., 2015)."

Introduction

Line 48: It is not accurate to say RA is the second largest source of carbon fluxes from soil because we don't know whether Ra is larger than Rh. And does the (Raich and Schlesinger 192) paper really say that? And in line 309 you said Rh account for 0.54-0.63, means RH > RA.

Response: we apologize for the improper statement. We mean soil respiration is the second largest carbon flux. We revise the text:

"RA is one main component of soil respiration (Hanson et al., 2000), and soil respiration represents the second largest source of carbon fluxes from soil to the atmosphere (after gross primary production, GPP) in the global carbon cycle (Raich and Schlesinger, 1992)."

Line 54: there is a new study summarized global Rs estimates: Bond-Lamberty, Ben. "New techniques and data for understanding the global soil respiration flux." Earth's Future 6.9 (2018): 1176-1180.

Response: thank you for your recommendation. We cited the global estimates of soil respiration summarized by Bond-Lamberty (2018) to support our study:

"RA could amount roughly up to 54 Pg C yr-1 (1 Pg = $10^{15}$ g, calculating RA as an approximate ratio of 0.5 of soil respiration, more details in Hanson et al., 2000) according to different estimates of global soil respiration (Bond-Lamberty, 2018), which is almost 5 times of the carbon release from human activities (Le Quéré et al., 2018)."

Line 62-63: a citation needs to support this statement.

Response: done! We revised the text as follows:

"Although individual site measurements of RA have been widely conducted across ecosystem types and

biomes, the globally spatial and temporal patterns of RA have not been explored and still act as a "black box" in global carbon cycling (Ballantyne et al., 2017)."

Line 63-64: need a citation.

Response: Revised as follows in the text:

"This "black box" is not well constrained and validated, because most terrestrial ecosystem models and earth system models were commonly calibrated and validated against eddy covariance measurements of net ecosystem carbon exchange (Yang et al., 2013)."

Line 85: "linear of non-linear models" change to "linear and non-linear models".

Response: done!

Line 86: But in line 94, you said RF model can avoid overfitting. Zhao et al 2017 used ANN models; and Jian et al 2018 also include RF models. So you need to be concise to avoid inconsistent.

Response: Zhao et al 2017 was appropriate and removed in L94!

Line 95: Zhao et al. 2017 used ANN models, it is not appropriate to cite here.

Response: Zhao et al. 2017 was removed, while Bodesheim et al., 2018 and Jung et al. 2017 were cited here.

Line 96: It is better also include the GitHub commit number of SRDB.

Response: as suggested the editor, we cited a more updated citation, which is from https://daac.ornl.gov/cgi-bin/dsviewer.pl?ds_id=1578 (Bond-Lamberty and Thomson, 2018). Therefore, we included version number here.

"First, RA observational dataset was developed based on SRDB (v4) across the globe, which is publicly available at https://daac.ornl.gov/cgi-bin/dsviewer.pl?ds_id=1578 (Bond-Lamberty and Thomson, 2018)."

Line 105: other environmental factors is too broad, please to be more specific.

Response: revised! We specified the soil and vegetation factors.

"which further advance our knowledge of the co-variation of RA with climate, soil and vegetation factors"

Material and methods

A big point in this study is you compared your results with that from Hashimoto (2015), you need to talk

about how you get the RA data of Hashimoto (2015). You directly used their data or you reproduced their estimates. If you reproduced, how and whether you used the same climate data as Hashimoto?

Response: we apologize for the misleading of Hashimoto RA. We first added a short summary about the details of input data, the method used and how soil respiration and RA were derived to avoid that the readers of this manuscript need to read Hashimoto et al. 2015 to understand the RA dataset. Then we provided details on how to get Hashimoto2015-RA (renamed as suggested by editor to fit ESSD better).

Detailed in text as:

"Hashimoto developed a climate-driven model by updating Raich's model, which stimulated soil respiration as a function of temperature and water (precipitation) at a monthly time step (Hashimoto et al., 2015; Raich et al., 2002). Therefore, to get a global estimate to soil respiration at a monthly scale, the globally gridded air temperature and precipitation with a spatial resolution of 0.5° were derived from University of East Anglia CRU 3.21 (Harris et al., 2014), and 1638 field observations were taken from SRDB (v3) for model parameterization (Hashimoto et al., 2015). Monthly soil respiration was summed to a yearly scale. Furthermore, annual soil respiration was divided into autotrophic and heterotrophic respiration using a global relationship between soil respiration and heterotrophic respiration derived from a meta-analysis (Bond-Lamberty et al., 2004). This global relationship can be expressed by:

$$\ln(RH) = 1.22 + 1.73 \times \ln(RS) \qquad (1)$$

Where RH means annual heterotrophic respiration, and RS stands for annual soil respiration, expressed by g C m$^{-2}$ yr$^{-1}$.

Therefore, global Hashimoto2015-RA was derived by the difference between soil respiration and heterotrophic respiration. The monthly or annual Hashimoto2015-RA dataset can be freely accessed from (http://cse.ffpri.affrc.go.jp/shojih/data/index.html, Hashimoto et al., 2015)."

Line 110-112: are those papers from CNKI all in Chinese? How many studies and how many more data records you got from that? Please clarify that.

Response: yes, those papers from CNKI are all in Chinese with English abstract. We added 68 more RA observations and revised in the text:

"Finally, this study included a total of 449 field observations (Fig. 1), including 68 observations from CNKI."

Line 122: Australia, Russia, Africa, and South America.

Response: done!

Line 145: The srdb v4 covered 1960-2017, why your study only covered 1980-2012?

Results: from 1960 to 1980, there are only 11 observations, which might bring uncertainties. Our study covered the period until 2012 for easily comparing with Hashimoto RA, which covered the period up to 2012.

Line 224: '-4 – 4' change to '-4 to 4'.

Response: done!

Line 224-225: 'East Russia and tropical and Eastern regions in Africa' change to 'East Russia, tropical, and Eastern regions in Africa'.

Response: done!

Line 264-265: Usually anomaly was the difference between temperature/precipitation of corresponding year to the mean of a period (e.g., 1980-2012 in this study). But this should not change the results, if previous studies calculate anomaly like yours, please provide a citation to support.

Response: thank you for your suggestion, and we followed the suggestion. The anomaly of temperature/precipitation of corresponding year to the mean of 1980-2012, and the results did not change.

Line 270-273: why in temperate zone/savannas/wetland there is no correlation between RA and temperature anomaly? That is interesting, usually, in tropical and subtropical regions, Rs is less correlated with temperature (and should be also true for the temperature anomaly). I think it worth to analyze in more details and try to explain the mechanism or maybe just because of the uncertainty.

Response: the different responses of ecosystem types or climate zones to climatic variables may be related to regional heterogeneity and plant functional trait. For example, regional temperature significantly differed from global averages (Huang et al., 2012), with much faster change in high-latitude regions (Hartmann et al., 2014), and semi-arid dominated the trend and variability of global land $CO_2$ sink (Ahlström et al., 2015). Similar studies were also found in other studies, e.g. total soil respiration or RA (Ballantyne et al., 2017; Jian et al., 2018a). Therefore, the regionally uneven responses of RA to climatic variables were unlikely due to model uncertainty.

These results have been discussed in "**4.1 Global RA**" section, and we revised the text as:

"However, RA increment varied with climate zones and ecosystem types (Figs. S2 and S3), which was similar to previous findings that total soil respiration or RA varied with climate zones or ecosystem types (Ballantyne et al., 2017; Jian et al., 2018a). These differences may be related to regional heterogeneity and plant functional trait. For example, regional temperature significantly differed from global averages (Huang et al., 2012), with much faster change in high-latitude regions (Hartmann et al., 2014), and semi-arid dominated the trend and variability of global land $CO_2$ sink (Ahlström et al., 2015)."

Line 310-311: See also Lamberty 2018 Earth's Future paper. "New techniques and data for understanding the global soil respiration flux." Earth's Future 6.9 (2018): 1176- 1180.

Response: thank you for the recommendation, and we cited the global soil respiration estimates from Bond-Lamberty (2018):

"Bond-Lamberty et al. (2018) proposed that the global average proportion of heterotrophic respiration ranged from 0.54 to 0.63 over 1990-2014 and global soil respiration was 67 to 108 Pg C yr$^{-1}$ according to different estimates Bond-Lamberty (2018); (Bond-Lamberty and Thomson, 2010b; Hashimoto et al., 2015; Hursh et al., 2017; Jian et al., 2018b), thus global RA varied from 25 to 51 Pg C yr$^{-1}$."

Discussion

Dominant factors: all you talked were about driving factors of RA spatial variability, right? Did you also analyze the dominant factors of temporal variability? Limitation and uncertainty: see my previous overall comment. In addition, Jian et al. "Constraining estimates of global soil respiration by quantifying sources of variability." Global change biology 24.9 (2018): 4143-4159 talked about uncertainty related to time-scaling and Rs upscaling. How about RA upscaling and timescale?

Response: we analyzed the dominate factors at both spatial and temporal patterns. We used partial correlation analysis based on a timescale from 1980 to 2012 for each grid cell (see methodology section 2.5), and the correlation coefficient was applied to derive the dominant factor map (Fig. 9). However, we did not analyze the dominant factors for each given year.

We additionally discussed the potential variability of RA using different time scale variables in "**4.4 Advantages, limitations and uncertainties**".

"Second, due to the limited observations of RA at a daily or monthly scale, the RF-RA dataset was

produced at an annual scale. Although there was no direct study to compare the difference of RA upscaling from daily or monthly and annual scale, substantial differences of soil respiration upscaling from daily or monthly and annual scales (Jian et al., 2018b) indirectly illustrated the potential difference of RA upscaling from different timescales."

Author contributions

Line 445: 'to the review the manuscript' change to 'to review the manuscript'.

Response: done.

[revised manuscript text omitted]

| | Downward Shortwave radiation | Split | Yearly | https://www.esrl.noaa.gov |
| | Nitrogen deposition | Split | Yearly | https://www.isimip.org/gettingstarted/availability-input-data-isimip2b/ |
| Soil | Soil carbon content | – | Static | https://soilgrids.org/#!/?layer=TAXNWRB_250m(Hengl et al. 2017) |
| | Soil nitrogen content | – | Static | https://webmap.ornl.gov/ogc/index.jsp |
| | Soil water content | Split | Yearly | https://www.esrl.noaa.gov/psd/data/gridded/data.cpcsoil.html |
| Vegetation | MODIS land cover | – | Static | https://glcf.umd.edu/data/lc/ |

[1]Although this study tried to link some variables relating to plant activities, such as Normalized Difference Vegetation Index (NDVI), Leaf Area Index (LAI), however, these variables could not help to improve the model efficiency. Due to the lack of fully land cover of these products, and the plant activities could be indirectly reflected by temperature, precipitation, potential evaportransporation, soil nutrients, etc., therefore, this study did not use NDVI or LAI for spatial and temporal modelling of RA.

Figures

[Figure]

**Fig. S1.** Comparison between data-derived belowground autotrophic respiration (RA) and observed RA using a 10-fold cross-validation.

[Figure]

**Fig. S2.** Inter-annual variability of belowground autotrophic respiration (RA) for RF-RA (a) and Hashimoto2015-RA (b) for boreal, temporal and tropical areas

[Figure]

**Fig. S3.** Inter-annual variability of belowground autotrophic respiration (RA) for RF-RA (a) and Hashimoto2015-RA (b) for boreal forest, cropland, grassland, savannas, shrubland, temperate forest, tropical forest and wetland.

[Figure]

**Fig. S4.** The relationships between total belowground autotrophic respiration (RA) and temperature/precipitation anomaly for RF-RARF-RA (a) and Hashimoto2015-RA (b) for boreal, temperate and tropical areas.

[Figure]

**Fig. S5.** The relationships between total belowground autotrophic respiration (RA) and temperature/precipitation anomaly for

RF-RA (a) and Hashimoto2015-RA (b) for eight biomes

[Figure]

**Fig. S6.** Latitudinal patterns of partial correlation coefficient between RF-RA and mean annual temperature (MAT), mean annual precipitation (MAP) and shortwave radiation (SWR).

[Figure]

**Fig. S7.** The percentage of dominant factor for global RA (calculated from cell areas).

[Figure]

**Fig. S8.**

---

## Author Response (AR3)

**Response to reviewer**

Dear reviewer,

Thank you again for your revision. According to your suggestion, I have revised the two minor comments. Please see below.

Best wishes,

Xiaolu Tang, on behalf of all co-authors.

(1) The citation at the end (i.e., Bond-Lamberty and Thomson, 2010b; 295 Hashimoto et al., 2015; Hursh et al., 2017; Jian et al., 2018b) should be deleted because you already cited in the beginning.

Response: done! "Bond-Lamberty and Thomson, 2010b; 295 Hashimoto et al., 2015; Hursh et al., 2017; Jian et al., 2018b" were removed.

(2) suggest 'average of heterotrophic respiration ranged from 0.54 to 0.63' change to 'average of autotrophic respiration ranged from 0.37 to 0.48'.

Response: done as:

"The global average proportion of RA ranged from 0.37 to 0.46 over 1990-2014 (calculated from Bond-Lamberty et al., 2018), while global soil respiration was 67 to 108 Pg C $yr^{-1}$ according to different estimates, thus global RA varied from 25 to 51 Pg C $yr^{-1}$."